# Microbial dynamics and pulmonary immune responses in COVID-19 secondary bacterial pneumonia

Natasha Spottiswoode [1,38], Alexandra Tsitsiklis[1,38], Victoria T. Chu [2,3], Hoang Van Phan[1], Catherine DeVoe [1], Christina Love[1], Rajani Ghale[1,4], Joshua Bloomstein [5], Beth Shoshana Zha [4], Cole P. Maguire [6], Abigail Glascock[3], Aartik Sarma [4], Peter M. Mourani [7], Katrina L. Kalantar[8], Angela Detweiler [3], Norma Neff [3], Sidney C. Haller[4], COMET Consortium*, Joseph L. DeRisi[3,9], David J. Erle [4,10,11], Carolyn M. Hendrickson [4], Kirsten N. Kangelaris[12], Matthew F. Krummel [13], Michael A. Matthay [4], Prescott G. Woodruff[4,11], Carolyn S. Calfee[4] & Charles R. Langelier [1,3] ✉

Secondary bacterial pneumonia (2°BP) is associated with significant morbidity following respiratory viral infection, yet remains incompletely understood. In a prospective cohort of 112 critically ill adults intubated for COVID-19, we comparatively assess longitudinal airway microbiome dynamics and the pulmonary transcriptome of patients who developed 2°BP versus controls who did not. We find that 2°BP is significantly associated with both mortality and corticosteroid treatment. The pulmonary microbiome in 2°BP is characterized by increased bacterial RNA mass and dominance of culture-confirmed pathogens, detectable days prior to 2°BP clinical diagnosis, and frequently also present in nasal swabs. Assessment of the pulmonary transcriptome reveals suppressed TNFα signaling in patients with 2°BP, and sensitivity analyses suggest this finding is mediated by corticosteroid treatment. Further, we find that increased bacterial RNA mass correlates with reduced expression of innate and adaptive immunity genes in both 2°BP patients and controls. Taken together, our findings provide fresh insights into the microbial dynamics and host immune features of COVID-19-associated 2°BP, and suggest that suppressed immune signaling, potentially mediated by corticosteroid treatment, permits expansion of opportunistic bacterial pathogens.

Secondary bacterial pneumonia (2°BP) is a morbid and often fatal complication of severe respiratory viral infections[1-6]. Hospital-acquired 2°BP has been especially problematic during the COVID-19 pandemic, leading to longer hospitalizations[3,6], increased mortality[7,8], and a rise in antimicrobial-resistant (AMR) infections[9,10]. A dynamic relationship between pathogens, the lung microbiome, and the host immune response underpins the pathophysiology of pneumonia[11,12], yet few studies have assessed these biological features in patients with 2°BP, leaving gaps in our understanding of this important sequela of viral illness.

The co-pathogenesis of respiratory viruses with bacterial pathogens has been recognized for decades, and best studied in the

A full list of affiliations appears at the end of the paper. *A list of authors and their affiliations appears at the end of the paper.
✉e-mail: chaz.langelier@ucsf.edu

context of influenza virus[13,14]. In the 1918 influenza pandemic, which led to over 50 million deaths, retrospective autopsy studies revealed evidence of 2°BP in the majority of cases[14]. Influenza, COVID-19, and other viral infections lead to alterations in the upper and lower respiratory tract microbiome, which may increase susceptibility to secondary infections by creating ecological niches for pathogenic bacteria[12,15]. Reduced microbiome alpha diversity, for instance, is a feature of both viral and bacterial lower respiratory tract infections in critically ill patients[12,16–18].

The common practice of early empiric antimicrobial administration in critically ill patients can further disrupt the airway microbiome, and additionally select for AMR bacterial pathogens[12,16]. Mechanically ventilated patients in particular endure prolonged exposure to the hospital environment, which increases the risk of colonization by opportunistic pathogens in the upper airway, oropharynx, and lungs[19,20]. It remains unclear, however, how changes in the airway microbiome of critically ill patients with COVID-19 might precipitate 2°BP, highlighting an important knowledge gap.

Severe COVID-19 is characterized by a dysregulated inflammatory response in both the airways and systemic circulation[21,22], yet whether 2°BP is associated with further alterations in this pathologic immune state remains unclear. For instance, 2°BP might lead to activation of innate immune signaling pathways important for bacterial defense, which has been observed in patients with ventilator-associated pneumonia prior to the COVID-19 pandemic[23–25]. Alternatively, 2°BP might arise from suppressed immune signaling, which is well described in mouse models of post-influenza 2°BP[13,26], and in patients with sepsis who acquire nosocomial infections[20,27]. It is also possible that the host response to 2°BP may simply be overshadowed by the inflammatory state of severe SARS-CoV-2 infection.

Despite their interconnected roles, few studies have assessed both the lower respiratory tract microbiome and host immune responses in critically ill patients, and none for the explicit purpose of studying post-viral 2°BP. A recent elegant study showcased how lower respiratory metatranscriptomics can effectively identify connections between host and microbial factors with clinical outcomes in COVID-19[28], however it did not focus on clinically confirmed 2°BP. Two recent diagnostic test studies demonstrated the potential of respiratory metatranscriptomics to improve the detection of pathogens in COVID-19 patients with ventilator associated pneumonia[29,30], but did not evaluate biological features of 2°BP.

The burden of secondary infections in patients with COVID-19 and other viral pneumonias, as well as gaps in our mechanistic understanding of 2°BP, motivated us to carry out this study. We assessed lung microbiome dynamics and host immune responses using metatranscriptomics in a large cohort of hospitalized COVID-19 patients with rigorous 2°BP adjudication by three physicians. We observed disruption of the lung microbiome in patients with 2°BP, characterized by increased bacterial RNA mass and dominance of culture-identified pathogens, as well as changes in host immune signaling involving genes important for bacterial defense. Together, our findings provide fresh insights into the biology of 2°BP, suggesting potential new therapeutic targets and approaches to 2°BP diagnosis.

## Results
### Patient cohort and pneumonia adjudication
We studied critically ill patients requiring mechanical ventilation for COVID-19 enrolled in the prospective observational COVID-19 Multi-Phenotyping for Effective Therapies (COMET) study between 04/2020 and 12/2021. Patients were enrolled at one tertiary care hospital and one safety net hospital in San Francisco, California under a research protocol approved by the University of California San Francisco Institutional Review Board (Methods). We collected tracheal aspirate (TA) and nasal swabs (NS) periodically following intubation, and performed metatranscriptomic sequencing (Fig. 1a).

Of the 397 patients with COVID-19 enrolled in COMET, 112 had critical illness requiring invasive mechanical ventilation. Culture-confirmed 2°BP was identified in 44 (39.3%) of these patients based on 3-physician adjudication using the US Centers for Disease Control and Prevention PNEU1 surveillance definition of pneumonia[31] and all available clinical data in the electronic medical record, blinded to metatranscriptomic results. Patients with no clinical evidence of bacterial pneumonia at any point during their hospitalization ($N = 41$, No-BP group) were also identified. 27 patients were excluded from further analysis; 22 of whom could not be confidently adjudicated into 2°BP or No-BP groups, and 5 with other reasons for exclusion including hospital transfer or intubation for reasons other than COVID-19.

Analysis of clinical and demographic data (Supplementary Table 1) demonstrated that hospital mortality was significantly greater in patients with 2°BP than in those without (47.7% vs 7.3%, $P < 0.0001$). Patients with 2°BP were also more likely to have received corticosteroids during their hospitalizations (97.7% vs 82.9%, $P = 0.026$). All patients received antibiotics during their hospitalizations, and total days of antibiotic therapy in the first week of hospitalization did not differ between groups. A minority of patients had received one or more SARS-CoV-2 vaccines prior to admission (9.1% vs 12.2%, $P = 0.61$); most patients were recruited prior to vaccine availability.

For metatranscriptomic analyses, we evaluated patients with TA samples that met baseline quality control metrics, and for 2°BP patients, those with samples available within a 5-day window of 2°BP clinical diagnosis (Fig. 1, Methods). This left 178 TA samples from 27 2°BP and 29 No-BP patients available for metatranscriptomic analysis (Supplementary Table 2). Within this analysis subgroup, *Staphylococcus aureus* was the most prevalent 2°BP pathogen identified by clinical bacterial cultures ($N = 10$, 37.0%) followed by *Pseudomonas aeruginosa* ($N = 6$, 22.2%) (Fig. 1b, Supplementary Data 1). Fifteen of the 27 patients with 2°BP also had NS samples collected and suitable for analysis.

### COVID-19-associated secondary bacterial pneumonia is characterized by higher lower airway bacterial RNA mass, pathogen dominance, and changes in the lung microbiome
We began metatranscriptomic analyses by comparatively assessing bacterial RNA mass in the lower respiratory tract microbiome of 2°BP patients versus No-BP controls. TA samples from 2°BP patients collected closest to date of clinical diagnosis had higher bacterial RNA mass compared to samples from No-BP controls obtained at comparable timepoints post-intubation ($P = 0.016$, Fig. 2a). We next assessed lung microbiome alpha diversity and found that while median Shannon Diversity Index (SDI) was lower in 2°BP patients compared to No-BP controls, it did not significantly differ ($P = 0.10$, Fig. 2b).

While in most patients the 2°BP pathogen was most abundant in the TA sample collected closest to the time of clinical 2°BP diagnosis, in 10/27 (37.0%) of patients, it was ranked highest by abundance in samples collected earlier or later. We thus repeated the alpha diversity analysis using the sample in which the 2°BP pathogen was highest ranked (as opposed to the sample closest to the date of 2°BP diagnosis), and found that SDI was significantly lower compared to No-BP controls ($P = 0.014$, Fig. 2c). Longitudinal assessment demonstrated that SDI decreased over time in both groups, and was consistently lower in patients with 2°BP (Supplementary Fig. 1).

Community composition of the lung microbiome in 2°BP versus No-BP patients did not differ based on either Bray Curtis index, which considers taxon abundance (Fig. 2d, $P = 0.06$ by PERMANOVA), or Jaccard index, which considers taxon presence/absence (Supplementary Fig. 2, $P = 0.09$). We considered that corticosteroid exposure, days of mechanical ventilation prior to sampling, SARS-CoV-2 viral load and bacterial mass could each be potential confounders or effect modifiers of our lung microbiome analyses. In sensitivity analyses, however, we did not observe significant differences when evaluating the impact of

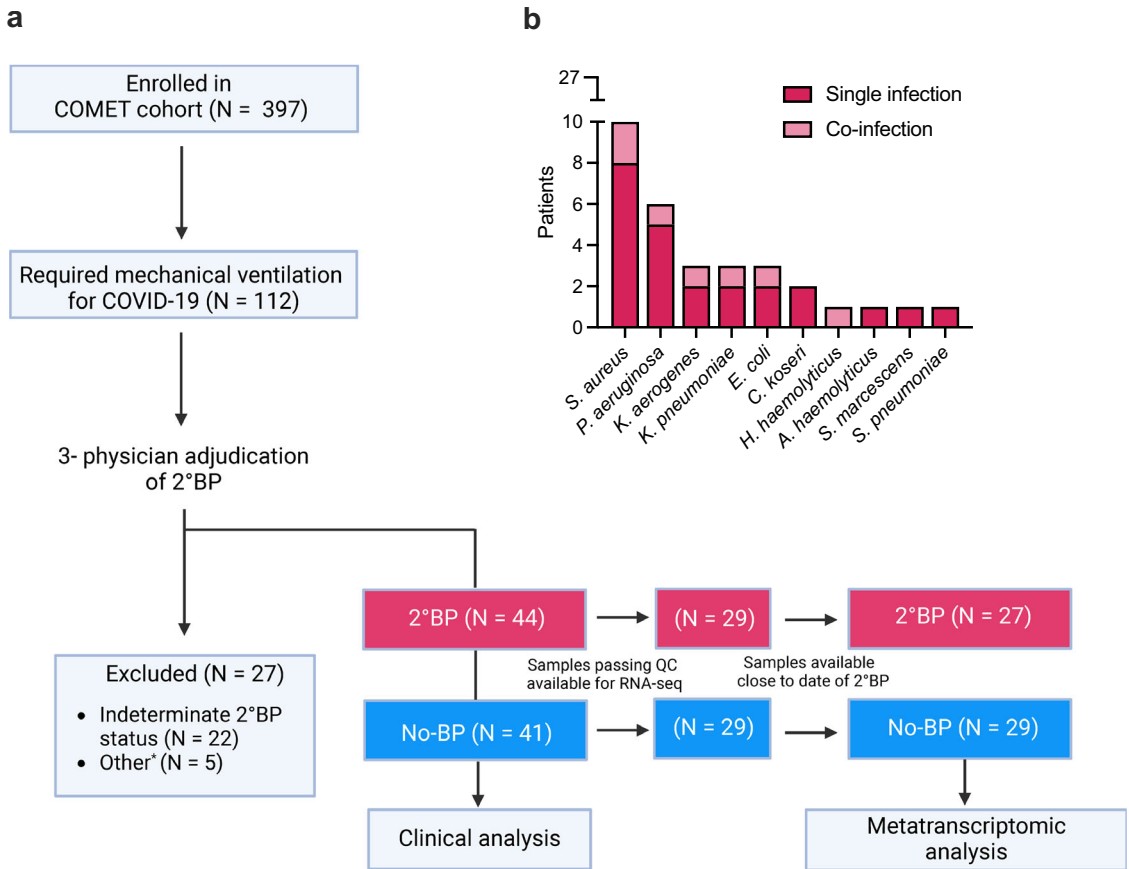

**Fig. 1 | Study overview. a** Patient flow after recruitment within 48 hours of admission. *Intubated for reasons other than COVID-19 or transferred to an outside hospital within 48 h of recruitment. **b** Pathogens detected by tracheal aspirate (TA) bacterial culture at time of secondary bacterial pneumonia (2°BP) diagnosis. Three patients had >1 pathogen detected. Created with BioRender.com released under a Creative Commons Attribution-NonCommercial-NoDerivs 4.0 International license (https://creativecommons.org/licenses/by-nc-nd/4.0/deed.en).

each of these variables on alpha diversity (Supplementary Fig. 3). Similarly, our beta diversity findings remained globally unchanged when adjusting for days of steroid exposure prior to sampling, days of mechanical ventilation, SARS-CoV-2 viral load or bacterial mass (Supplementary Fig. 4).

The culture-confirmed 2°BP pathogen was detected by metatranscriptomics in all 27 2°BP cases. Furthermore, it was dominant in the lower airway microbiome of at least one sample from most patients with 2°BP (Fig. 2e), with the culture-confirmed pathogen ranking in the top 3 most abundant taxa in at least one sample for 25/27 (92.6%) of cases within 7 days of clinical diagnosis (Fig. 2f). Further assessment of the most abundant taxa in lung microbiome demonstrated that *S. aureus* and *P. aeruginosa* were most commonly identified in 2°BP patients, while *Prevotella, Mycoplasma*, and *Alloprevotella* species were most commonly found in No-BP cases (Fig. 2g). Differential taxonomic abundance analysis demonstrated enrichment in *Klebsiella, Haemophilus, Citrobacter, Neisseria* and *Pseudomonas* species in 2°BP patients as compared to No-BP controls (Fig. 2h). A sensitivity analysis restricted to just patients who had received steroids demonstrated similar results (Supplementary Fig. 5).

We asked whether SARS-CoV-2 viral load in the lung, as measured by metatranscriptomics in reads per million (rpM), differed between patients with or without 2°BP, but found no difference ($P = 0.23$, Supplementary Fig. 6). Finally, we performed a functional analysis of bacterial metabolic pathways (see Methods). No significant differences in metabolic pathway abundances were observed at a false discovery

rate (FDR) < 0.1 between 2°BP and No-BP patients (Supplementary Fig. 7, Supplementary Data 2).

## Dynamics of secondary bacterial pneumonia pathogens in the lung

We next evaluated the longitudinal dynamics of the cultured-confirmed 2°BP pathogen in the lower airway based on abundance rank, pathogen mass, and pathogen mass normalized to total sample RNA mass, overlaid with phenotypic susceptibility to administered antibiotics (Fig. 3, Supplementary Figs. 8–10). This revealed cases where pathogen expansion in the lower airway prior to 2°BP diagnosis occurred in the absence of any antibiotic treatment (e.g., Patient 18, *Citrobacter koseri*, Fig. 3). We also observed cases where the abundance of the 2°BP pathogen decreased following clinical diagnosis (Patient 23, *Escherichia coli*, Fig. 3). Many pathogens, however, remained the most abundant microbe in the lower airway for days following initiation of antimicrobial therapy (e.g., Patient 16, *Pseudomonas aeruginosa*, Fig. 3), as well as following treatment with antibiotics to which the pathogen was resistant (e.g., Patient 13, *Klebsiella aerogenes*, Fig. 3). In patients with longitudinal sampling after clinical 2°BP diagnosis, we asked how pathogen clearance related to its susceptibility to administered antibiotics. We noted that patients with *Pseudomonas aeruginosa* infections were more likely to exhibit an impaired clearance phenotype compared to patients with other types of 2°BP ($P = 0.020$, Supplementary Fig. 11). Finally, we observed that the culture-confirmed 2°BP pathogen could be detected in the airway prior to

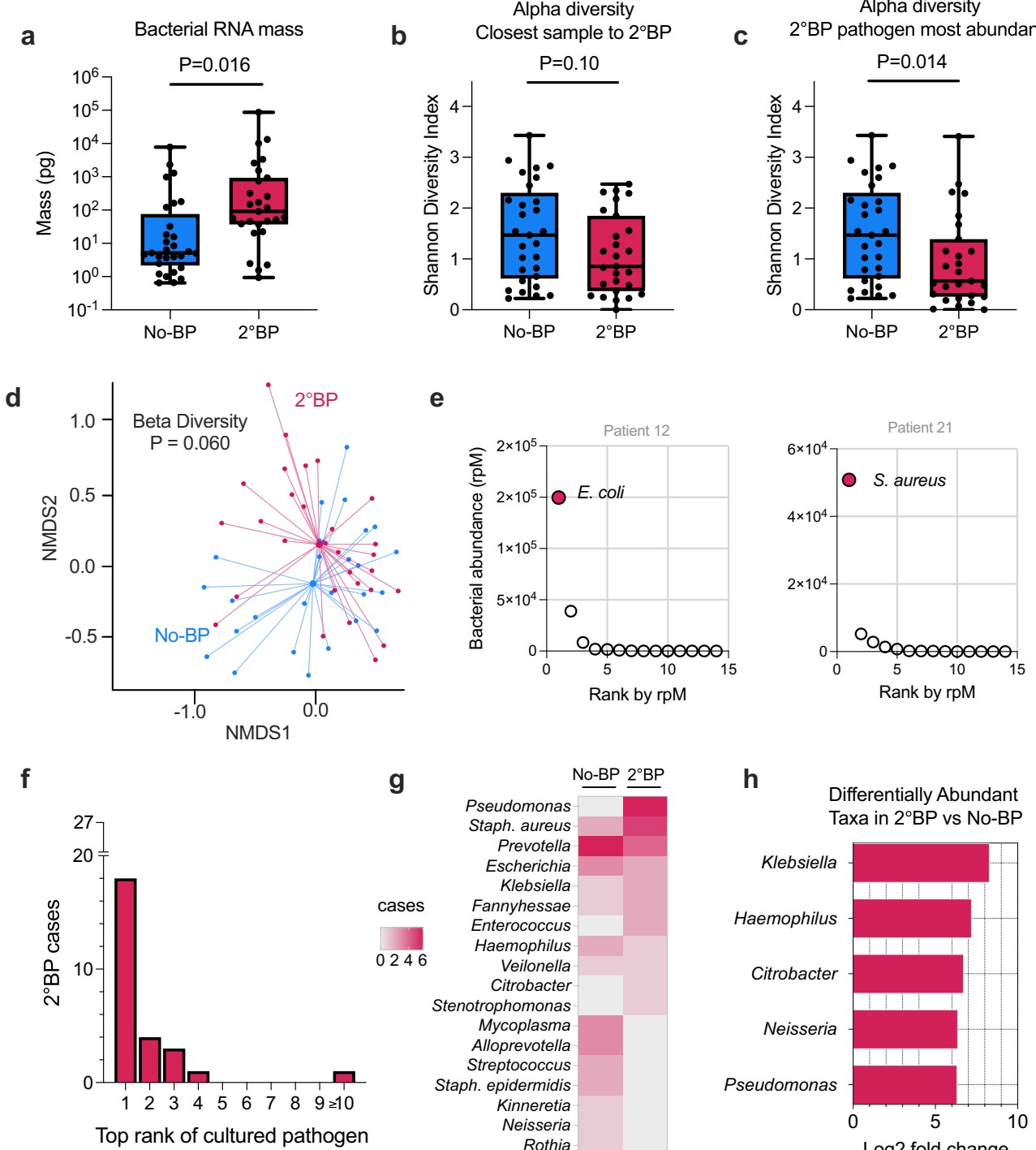

**Fig. 2 | Lung microbiome differences in patients with or without 2°BP.**
**a** Bacterial RNA mass and (**b**) alpha diversity measured by Shannon Diversity Index (SDI) in COVID-19 patients with 2°BP (red, *N* = 27) or No-BP (blue, *N* = 29), measured in the 2°BP TA sample closest to the date of clinical diagnosis, or No-BP patient samples obtained after a similar number of days post-intubation. **c** SDI measured in the 2°BP sample in which the culture-confirmed pathogen was top ranked by abundance in the lung microbiome (*N* = 27 patients with 2°BP, 29 with No-BP). **d** Compositional differences in the lung microbiome of 2°BP (red) versus No-BP (blue) patients based on Bray-Curtis dissimilarity index and the PERMANOVA test with 1000 permutations, visualized with a non-metric multidimensional scaling (NMDS) plot. **e** Exemplary abundance plots from two 2°BP patients demonstrating the abundance, measured in reads per million (rpM, y axis) for each of the top 15 ranked bacterial taxa in the lung microbiome (x axis). Culture-confirmed 2°BP pathogen highlighted in red. **f** Bar plot showing the number of 2°BP patients (y axis)

in which a specific rank by rpM of the culture-confirmed pathogen (x axis) was observed within 7 days of clinical diagnosis. **g** Heatmap showing the number of cases in which specific bacterial genera were detected as the most abundant in the bacterial lung microbiome, with the top 15 genera plotted. *Staphylococcus* is included at the species level given the intrinsic differences in potential pathogenicity between *S. aureus* and *S. epidermidis*. **h** Differentially abundant microbial genera in the lung microbiome of 2°BP versus No-BP patients. P values for (**a**–**c**) calculated by two-sided Wilcoxon test. Boxes in (**a**–**c**) show median and 25th–75th percentiles, with whiskers from min to max. P value in (**d**) calculated by two-sided PERMANOVA analysis. In (**h**), bacterial taxa with an adjusted *P* < 0.001 are plotted. A positive log fold change indicates enrichment for microbes in patients with 2°BP as compared to No-BP. *P* values in (**h**) are calculated by adjusted Wald's test. Source data for (**a**–**c**, **e**–**h**) provided in the Source Data file.

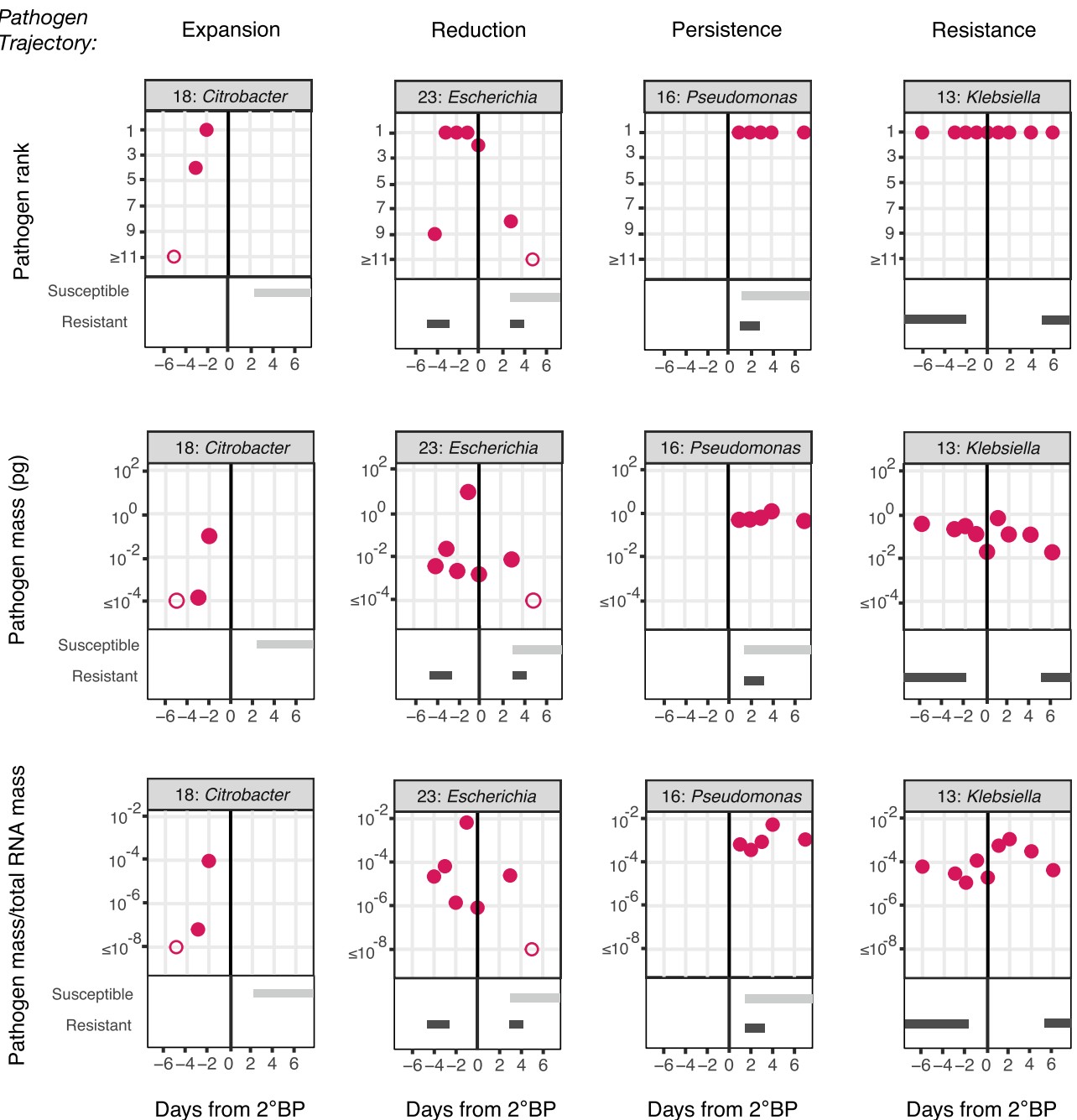

**Fig. 3 | Dynamics of 2°BP pathogen over time relative to the date of clinical diagnosis highlighting examples of unique pathogen trajectories.** The top row includes plots of genus level 2°BP pathogen rank based on bacterial reads per million (rpM) in the lung microbiome for each of four pathogen trajectories (expansion, reduction, persistence and resistance). The middle row includes plots of pathogen mass. The bottom row consists of plots of pathogen mass normalized to the total RNA mass of sample. Days relative to 2°BP clinical diagnosis are plotted on the X axis. Days during which patient received antibiotics to which 2°BP pathogen was phenotypically susceptible (light gray bar) or resistant (dark gray bar) are plotted below. The patient ID and pathogen genus are listed above each plot. Open circles denote samples in which the pathogen was not detected by metatranscriptomics.

2°BP diagnosis in 14/14 (100%) of the patients who had samples obtained prior to clinical diagnosis (Supplementary Figs. 8–10).

### Detection of secondary bacterial pneumonia pathogens in the upper respiratory tract

Among the 15 2°BP and 20 No-BP patients who also had nasal swab (NS) samples available, we did not observe differences in nasal microbiome bacterial RNA mass (Fig. 4a) or alpha diversity (Fig. 4b) based on 2°BP status. We noted that in 14/15 (93.3%) of cases with 2°BP, the 2°BP pathogen was detected in at least one NS sample within 7 days of

clinical diagnosis, and in 7/15 (46.7%) cases it ranked in the top 3 most abundant (Fig. 4c). Among the 9 patients who had NS samples collected prior to the date of clinical 2°BP diagnosis, 8/9 (88.9%) had at least one etiologic pathogen detected (Supplementary Figs. 12–14). We observed a low to moderate concordance between upper and lower respiratory tract microbiomes that did not significantly differ based on 2°BP status (median Spearman's rho 0.34, interquartile range (IQR) 0.13–0.54) for 2°BP patients versus 0.48 (IQR 0.26–0.61) for No-BP patients. Assessment of NS versus TA beta diversity based on Bray Curtis dissimilarity index also did not reveal significant differences

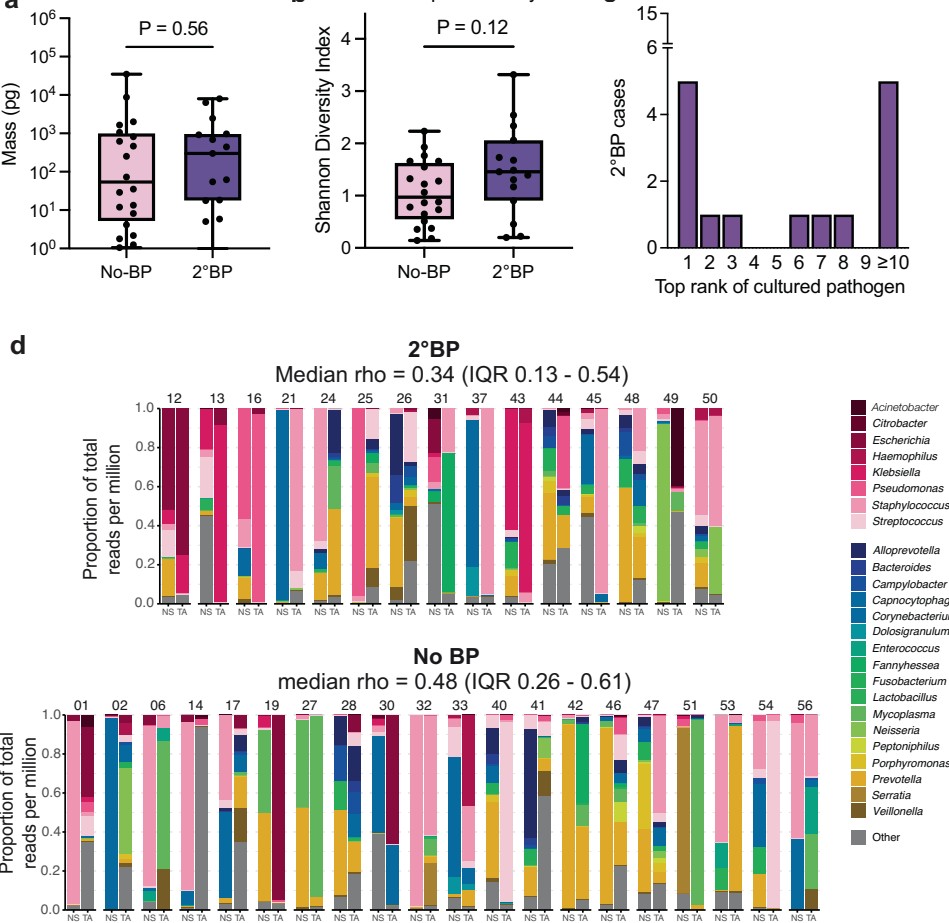

**Fig. 4 | Nasal microbiome differences in patients with or without 2°BP and relationship to the lung microbiome. a** Nasal swab (NS) bacterial RNA mass or (**b**) Shannon Diversity Index (SDI) in COVID-19 patients with 2°BP (purple, *N* = 15) or No-BP (pink, *N* = 20). *P* values based on two-sided Wilcoxon tests. **c** Bar plot showing the number of 2°BP patients (y axis) in which a specific rank by reads per million (rpM) of the culture-confirmed pathogen (x axis) was observed within 7 days of clinical diagnosis. **d** Stacked bar plots highlighting the taxonomic composition of paired NS and tracheal aspirate (TA) samples from 2°BP patients (top) and No-BP patients (bottom). Bacterial taxa present at <0.5% relative abundance across all samples were included in the "other" category, except if the microbe had been cultured as a 2°BP pathogen. Spearman's rho tests were performed to assess taxonomic concordance between paired samples from each patient. IQR, inter-quartile range. Boxes in (**a**, **b**) show median and 25th–75th percentiles, with whiskers from min to max. Source data for (**a**–**c**) provided in the Source Data file.

based on sampling site in 2°BP or No-BP patients (Supplementary Fig. 15).

## The respiratory antimicrobial resistome

Analysis of the lower respiratory tract resistome of 2°BP patients revealed a diversity of AMR genes representing multiple different classes, including plasmid-transmissible extended spectrum beta lactamase (ESBL) and colistin-resistance genes (e.g., *CTX-M* and *MCR-1*, respectively) (Fig. 5). In some cases, AMR genes associated with culture-confirmed resistant pathogens were detectable before clinical diagnosis of 2°BP (e.g., *CTX-M, SED-1*, patient 20) (Fig. 5, Supplementary Fig. 16). The inducible beta lactamase gene *ampC* was identified in patient 13, from whom *K. aerogenes* was identified by culture (Fig. 5). This pathogen continued to dominate the airway despite treatment with a beta-lactam antibiotic (piperacillin-tazobactam) to which the organism was phenotypically susceptible (Fig. 3). *AmpC* can be induced following beta lactam exposure in certain *Enterobacteriaceae*, resulting in reversal of phenotypic susceptibility, and in some cases clinical treatment failure[32]. We noted that in patients with *P. aeruginosa* 2°BP, 83% had *OXA-50, CatB7* and *Aph3-IIb*, detected, which confer resistance to beta-lactams, chloramphenicol and aminoglycosides, respectively (Supplementary Fig. 17).

Comparative assessment demonstrated that clinically relevant AMR genes detected in lower respiratory tract pathogens could also be identified in the upper airway in a subset of cases. For instance, in patient 12 (Fig. 5), who had multi-drug-resistant *E. coli, K. pneumoniae*, and methicillin-resistant *S. aureus* (MRSA) in TA culture, *MCR-1, CTX-M* and *mecA* were detected in both NS and TA samples. As in the lower respiratory tract, we found that clinically relevant AMR genes related to the 2°BP pathogen could in some cases be detected in the nares prior to clinical recognition of bacterial pneumonia (e.g., patient 12, Supplementary Fig. 16).

We additionally performed an exploratory genotype-to-phenotype analysis focusing on *mecA* and resistance to methicillin (and related beta lactams) in *S. aureus*, and *CTX-M* and resistance to ceftriaxone in *Enterobacteriaceae* isolates. We found that detection of *mecA* in TA within 7 days of 2°BP clinical diagnosis had a sensitivity of 100% (95% CI 18–100%), specificity of 67% (95% CI 30–94%), negative predictive value (NPV) of 100% (95% CI 51–100%) and positive predictive value (PPV) of 50% (95% CI 9–91%) for MRSA (Supplementary Table 3). Similar results were found with NS samples, which had a NPV of 100% (95% CI 51–100%) and a PPV of 67% (95% CI 12–98%) for MRSA pneumonia. With respect to ceftriaxone resistance in *Enterobacteriaceae*, we found

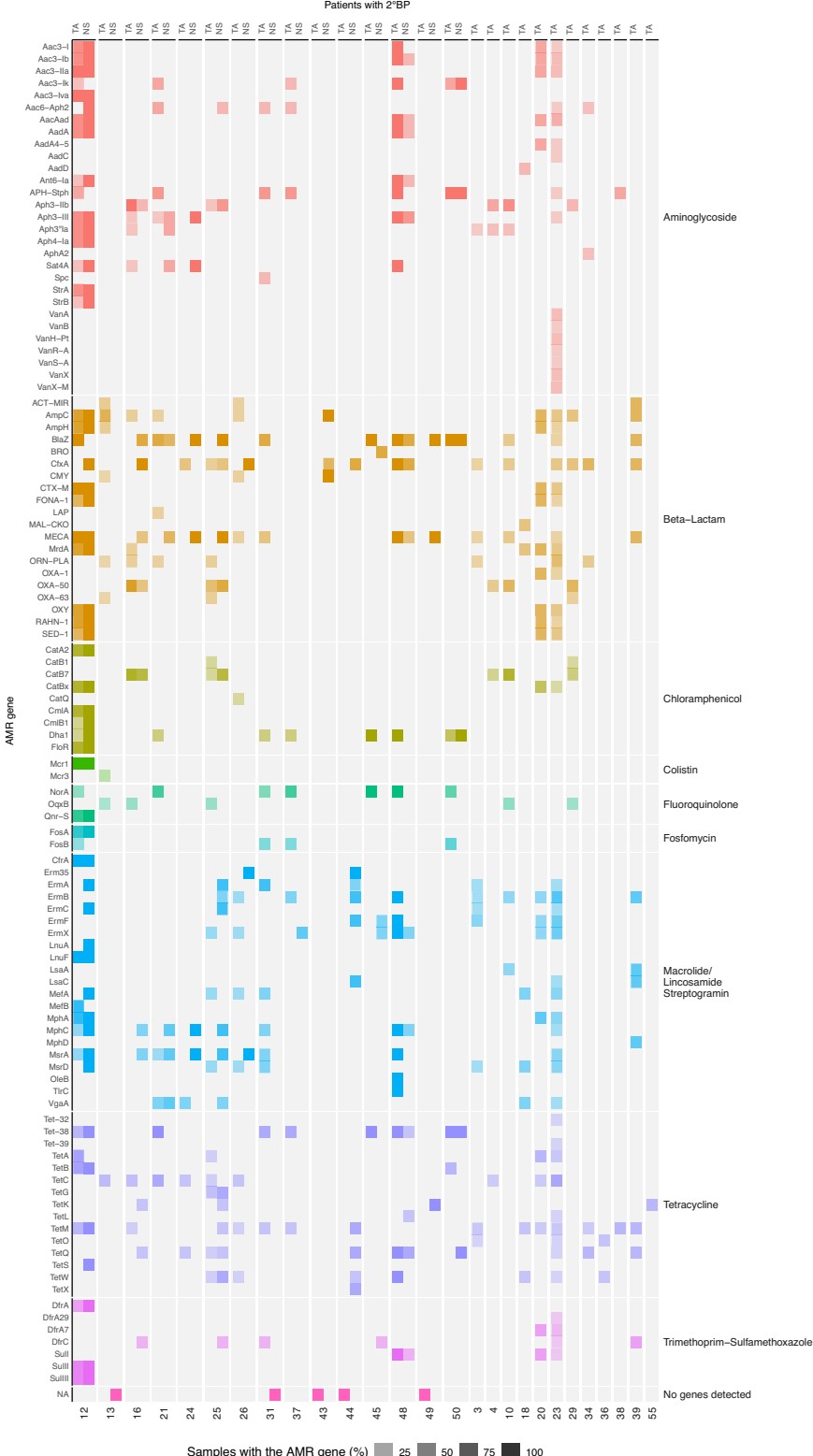

**Fig. 5 | Antimicrobial resistance (AMR) genes in the lung and nasal microbiome of 2°BP patients.** AMR genes, grouped and colored by class, detected in tracheal aspirate (TA) +/- matched nasal swab (NS) samples. Shading corresponds to the fraction of samples, collected within 7 days of 2°BP, in which the AMR gene was detected.

that detection of the ESBL gene *CTX-M* in TA had a sensitivity of 67% (95% CI 30–94%), specificity of 100% (95% CI 57–100%), NPV of 71% (95% CI 36–95%) and PPV of 100% (95% CI 51–100%) (Supplementary Table 4). In nasal swab samples, *CTX-M* had a NPV of 100% (95% CI 18–100%) and PPV of 100% (95% CI 5.1–100%) for

ceftriaxone resistance. We found that detection concordance between NS and TA samples varied based on AMR gene and patient (Supplementary Table 5). Finally, we tested for associations between specific AMR genes and in-hospital mortality, but found no significant associations (Supplementary Table 6).

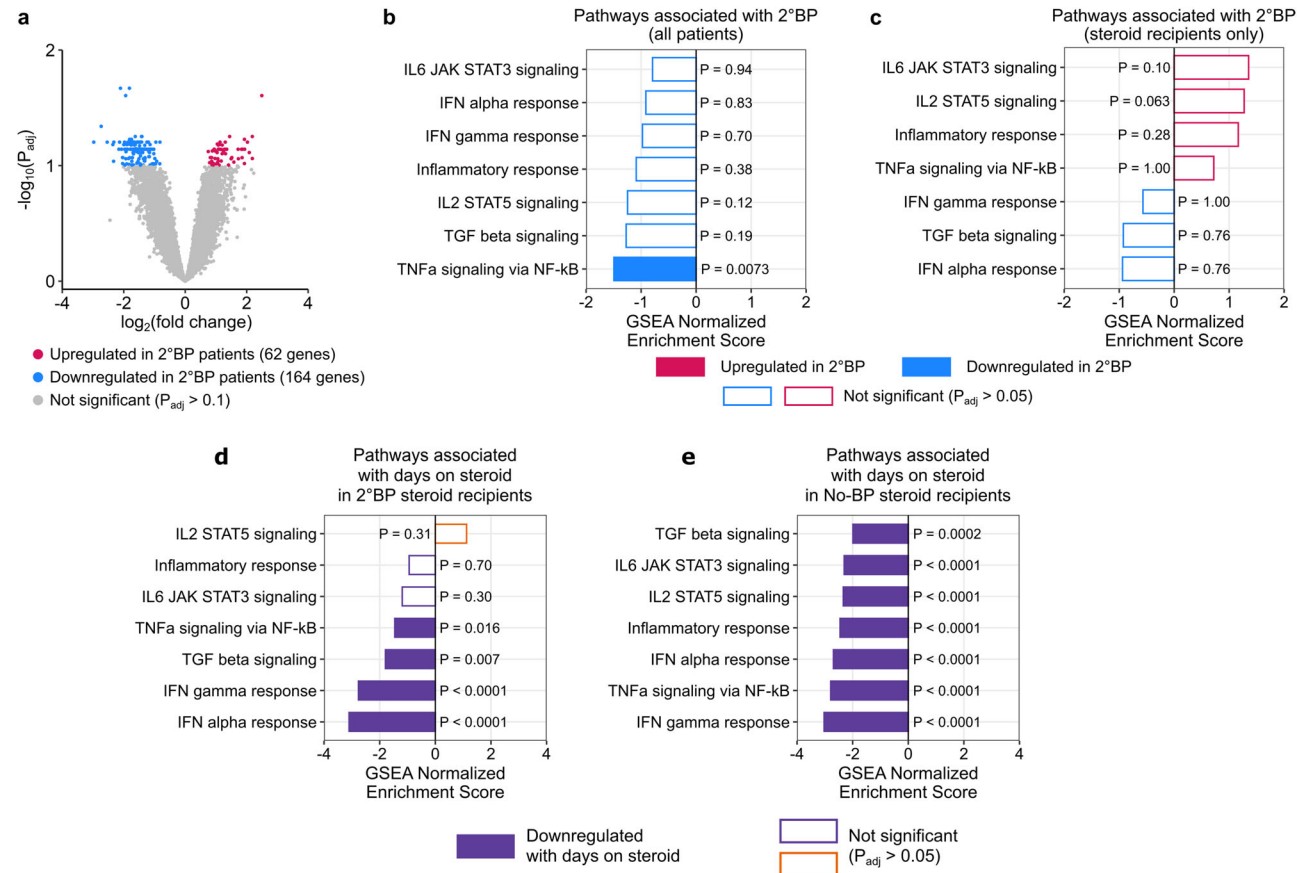

**Fig. 6 | Lower respiratory tract gene expression differs based on 2°BP status and is influenced by corticosteroid treatment. a** Volcano plot of differentially expressed genes between 2°BP ($N = 27$) and No-BP ($N = 29$). **b** Bar plot of GSEA analysis showing the Hallmark pathways that are downregulated in 2°BP patients. **c** Bar plot of GSEA analysis limited to steroid recipients ($N = 25$ 2°BP patients; $N = 19$ No-BP patients) showing the same Hallmark pathways as in (**b**). In the analyses in (**a**–**c**), we controlled for SARS-CoV-2 viral load in our differential expression analysis. **d** Bar plot of GSEA analysis demonstrating Hallmark pathways associated with days of corticosteroid treatment in 2°BP patients. **e** Bar plot of GSEA analysis demonstrating Hallmark pathways associated with days of corticosteroid treatment in No-BP patients. The two-sided $P$ values in (**a**) were calculated using linear modeling (limma package) and Benjamini-Hochberg correction. The two-sided $P$ values in (**b**–**e**) were calculated using the fgsea package and Benjamini-Hochberg correction.

## Host transcriptional responses in COVID-19 secondary bacterial pneumonia

We next asked whether a lower respiratory transcriptional signature of 2°BP could be identified amidst the intense inflammatory state of severe COVID-19. A comparison of host gene expression between 2°BP and No-BP patients identified 226 differentially expressed genes (FDR < 0.1) (Fig. 6a, Supplementary Data 3). Gene set enrichment analysis (GSEA) revealed that 2°BP was characterized by downregulated TNFα signaling via NF-κB in (Fig. 6b, Supplementary Data 4), suggesting that a state of suppressed antibacterial defense might characterize 2°BP in COVID-19 patients.

We hypothesized that corticosteroid treatment might be contributing to the observed relative suppression of immune responses in 2°BP patients. We thus performed a secondary differential gene expression analysis limited to only patients treated with dexamethasone or other corticosteroids prior to sampling ($N = 25$ 2°BP, $N = 19$ No-BP patients). This analysis yielded 425 differentially expressed genes (FDR < 0.1) (Supplementary Data 5), but no evidence of suppressed TNFα signaling by GSEA, supporting the idea that corticosteroid treatment may have contributed to the suppressed immune signaling observed in patients who developed 2°BP (Fig. 6c, Supplementary Data 6).

We sought to further understand the impact of steroid exposure on our transcriptomic findings by asking whether the number of days of steroid treatment prior to sampling influenced gene expression. As expected, we found that days of steroid exposure correlated with

suppression of several immune signaling pathways, including TNFα signaling, in both the 2°BP (Fig. 6d, Supplementary Data 7) and No-BP (Fig. 6e, Supplementary Data 8) groups. Together, these results suggested that a state of impaired immune defense, influenced by corticosteroid treatment, may exist in COVID-19 patients who develop 2°BP.

## Relative host immunosuppression is correlated with greater bacterial burden in the lungs

Finally, given the higher bacterial mass observed in 2°BP patients (Fig. 2a), and prior mouse studies suggesting that altered host gene expression in the context of experimental influenza virus infection is associated with bacterial overgrowth[26,33], we further investigated connections between the airway transcriptome and microbiome in 2°BP patients by testing whether bacterial RNA mass correlated with host gene expression. Using bacterial RNA mass as a continuous variable, differential expression analysis identified 4784 significant genes (FDR < 0.1) (Fig. 7a, Supplementary Data 9), and revealed an inverse relationship between bacterial mass and the expression of several innate and adaptive immunity genes (e.g., *HLA-DRB1*, *C1QC*) (Fig. 7b). A similar differential expression analysis carried out in No-BP patients, who had lower bacterial RNA mass (Fig. 2a), yielded 30 significant genes (Fig. 7c, Supplementary Data 10).

GSEA confirmed a marked inverse relationship between bacterial RNA mass and immune signaling pathways important for

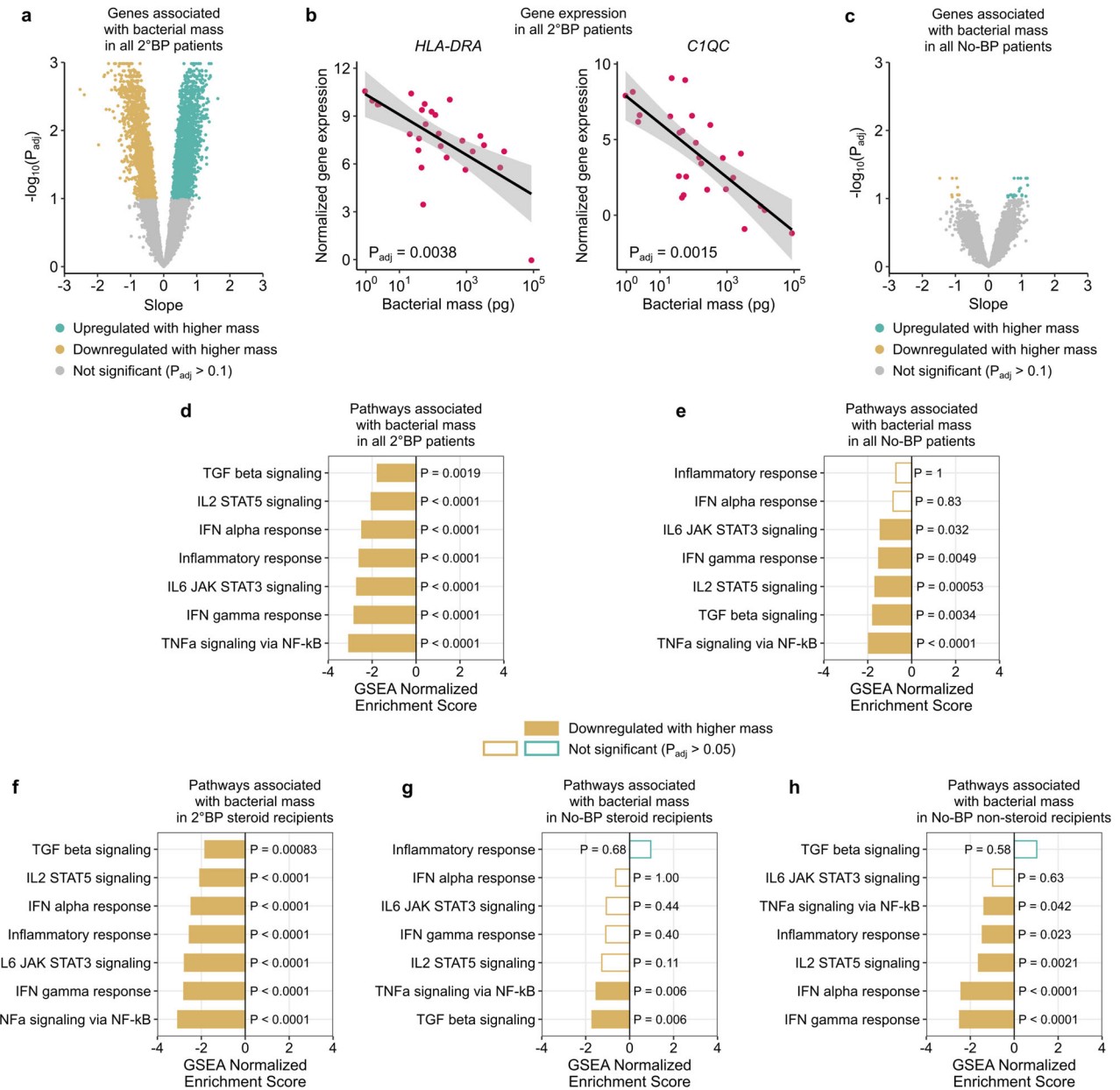

**Fig. 7 | Lower respiratory tract immune gene expression inversely correlates with bacterial mass. a** Volcano plots of genes that are associated with bacterial mass in 2°BP patients ($N = 27$). **b** Scatter plots showing the relationship between *HLA-DRA* and *C1QC* gene expression and bacterial RNA mass in 2°BP patients. The black lines indicate the linear regression fit, and the ribbons indicate the 95% confidence interval of the fits. **c** Volcano plots of genes associated with bacterial mass in No-BP patients ($N = 29$). Bar plot showing Hallmark pathways associated with bacterial mass in (**d**) all 2°BP patients and (**e**) all No-BP patients. Hallmark pathways associated with bacterial mass in (**f**) 2°BP and (**g**) No-BP patients, but limited to only steroid recipients ($N = 25$ and $N = 19$; respectively). **h** Hallmark pathways associated with bacterial mass in the 10 No-BP patients who did not receive steroids prior to sample collection. The two-sided $P$ values in (**a**, **c**) were calculated using linear modeling (limma package) and Benjamini-Hochberg correction. The two-sided $P$ values in (**d**–**h**) were calculated using the fgsea package and Benjamini-Hochberg correction.

antibacterial defense (e.g., TNFα, IL-6, IL-2) in both 2°BP (Fig. 7d, Supplementary Data 11) and No-BP patients (Fig. 7e, Supplementary Data 12). Restricting analyses to only 2°BP or No-BP patients who had received corticosteroids (Fig. 7f, g, and Supplementary Data 13 and 14) did not markedly change the relationship between bacterial mass and suppressed immune signaling, nor did restricting to the 10 No-BP patients without steroid exposure (Fig. 7h). Taken together, these results demonstrated that greater bacterial RNA mass was associated with lower expression of innate and adaptive immunity genes in mechanically ventilated COVID-19 patients, regardless of steroid exposure.

## Discussion

In this prospective observational study, we assessed respiratory tract microbial dynamics and host transcriptional responses associated with 2°BP in COVID-19 patients requiring invasive mechanical ventilation. Using comparative metatranscriptomics, we found that 2°BP is characterized by changes in the lung microbiome and host transcriptome, as well as dominance of pathogenic bacteria in the lungs that in many cases could be co-detected in the upper airway.

Bacterial superinfection is a well-established contributor to influenza mortality[14], yet in COVID-19 its role in mortality has been less clear[28,34,35]. We found that 2°BP affected 39.3% of mechanically

ventilated patients in our multicenter cohort, and was strongly associated with mortality. Our results notably differed from two important prior studies[28,35] which did not find clear links between secondary bacterial infection and mortality in COVID-19 patients. This discrepancy might be explained by our use of both an established case definition[31] and microbiological criteria to define 2°BP, as opposed to only requiring a positive bacterial respiratory culture[28,35]. We also found that patients who developed 2°BP were more likely to have received corticosteroids during their hospitalizations, suggesting that the therapeutic benefit of corticosteroids may come at the expense of increased 2°BP risk.

Early identification and treatment of hospital-onset bacterial pneumonia can prevent adverse consequences including prolonged mechanical ventilation, inappropriate antibiotic exposure, and mortality[36–38]. We found that 2°BP pathogens could be detected in the lower airway up to a week before clinical recognition of infection, and were frequently amongst the most abundant taxa in the lung microbiome in the days preceding culture-based detection. In some cases, we also detected pathogen-associated AMR genes before clinical diagnosis of 2°BP. Further studies in a larger, appropriately designed cohort, will be needed to ascertain the diagnostic performance of metatranscriptomics for 2°BP.

Lower respiratory tract infections are characterized by a loss of airway microbiome alpha diversity[12,16,39,40], which is also observed over time in mechanically ventilated patients, including those with COVID-19[41–44]. We found that 2°BP further disrupts microbial community structure in patients with existing SARS-CoV-2 infection, leading to additional loss of diversity in the setting of bacterial pathogen dominance in the airway. In over a third of cases, however, the peak of bacterial pathogen dominance, and consequently the greatest reduction in alpha diversity, did not overlap with the date of 2°BP clinical diagnosis. This could represent a decoupling of physiologic responses and pathogen dynamics, heterogeneity in TA sampling, or reflect the challenge of identifying a new pneumonia amidst an existing severe viral lower respiratory tract infection. By characterizing relationships between 2°BP pathogens and the lung microbiome over time, we observed that in some patients, the pathogen remained dominant in the airway even after 2°BP clinical diagnosis and the initiation of appropriate antibiotics. In particular, all *P. aeruginosa* cases exhibited this persistence trend, which may reflect the known tendency of this pathogen to form biofilms.

We found that 2°BP pathogens were not only detectable in the lower airway, but also in the upper airway in more than half of the cases, including prior to clinical pneumonia diagnosis. More broadly, however, we found only low-moderate correlation between the nasal and lower respiratory tract microbiomes, and did not observe any differences based on 2°BP status. A prior study using 16S rRNA gene sequencing to profile the microbiome of paired nasal and lower respiratory tract samples from children found significant correlations in taxonomic abundance between sites[45]. In contrast, a recent metatranscriptomic study of mechanically ventilated COVID-19 patients reported significant differences in the microbial composition of the upper and lower airways[28], as did a 16S study comparing the lower airway microbiome to that of the oropharynx and upper respiratory tract[46]. Additional studies are needed to more comprehensively investigate the relationships between the upper and lower airway microbiome, and how they differ based on sequencing technique (16S, metagenomics and metatranscriptomics) or on the presence of bacterial pneumonia.

AMR infections, in particular respiratory infections, are a major public health issue and increased in prevalence during the COVID-19 pandemic[9,10]. We identified several AMR genes considered potential threats to the management of bacterial infections in the lower airway metatranscriptomes of 2°BP patients. These included the plasmid transmissible genes *MCR-1*, *Qnr-S* and *CTX-M*, which have a high potential for horizontal transfer both within patient and within hospital. We also identified several AMR genes not typically detectable by existing clinical assays. For instance, three patients from one study site, who had temporospatial overlap during their hospital admissions, were found to harbor the *SED-1* class A beta lactamase (patients 12, 20, and 23). *SED-1* was originally identified in *Citrobacter sedikiae*[47], reported once in *E. coli* contaminating produce in China[48], but not otherwise reported in the context of human infection.

In several cases, pathogen-associated AMR genes were detectable before the clinical diagnosis of 2°BP, and were also found in the upper airway. Our genotype to phenotype assessment suggested that metatranscriptomic detection of clinically important AMR genes in NS or TA specimens may have utility for identifying patients with drug-resistant pneumonia. For instance, *mecA* detection in either TA or NS within 7 days of 2°BP clinical diagnosis had excellent sensitivity (100%) and NPV, and moderate specificity (67–80%) and PPV, for culture-confirmed MRSA pneumonia. In comparison, a meta-analysis of 22 studies found that *mecA* PCR had a pooled sensitivity and specificity of 70.9% and 90.3%, respectively[49]. We also found that *CTX-M* detection in either NS or TA had excellent PPV and moderate NPV for ceftriaxone resistance in *Enterobacteriaceae*. While these findings are promising and may have implications for both antimicrobial stewardship and infection control, a larger sample size is needed to draw robust conclusions.

Severe COVID-19 is characterized by a profoundly dysregulated host response in the lower respiratory tract[21,50,51], which complicates the assessment of a transcriptional immune response to 2°BP. Nonetheless, our results suggest that treatment with corticosteroids, now standard of care for severe COVID-19[52], drives suppression of innate immune signaling pathways important for antibacterial defense, such as TNFα signaling through NFκB. Indeed, increased rates of serious bacterial infections are well described in patients receiving anti-TNFα therapies[53], and our findings suggest that a state of relative immunosuppression mediated by corticosteroid treatment may augment susceptibility to opportunistic bacterial pathogens in COVID-19 patients.

Our observation that bacterial RNA mass in the lungs inversely correlates with the expression of immune signaling genes suggests that impaired antibacterial defense may enable the outgrowth of 2°BP pathogens. This idea is in line with results from murine studies of post-influenza 2°BP, which demonstrate virus-induced impairment of innate immunity, characterized by reduced expression of TNFα, IL-6, and other cytokines[13,26,54,55].

Strengths of our work include the use of host/microbe metatranscriptomics to study secondary bacterial infections, rigorous clinical adjudication of 2°BP, longitudinal sampling, and measurement of bacterial RNA mass, a biomarker not previously evaluated in studies of pneumonia. Limitations include a relatively small sample size and incomplete longitudinal sampling for all patients, which reduced the number of patients with analyzable samples near the date of 2°BP onset, and the number of analyzable longitudinal samples. This likely limited our ability to detect microbiome and host transcriptional differences that may have existed between 2°BP patients and controls. Our findings require further validation in independent cohorts. While several public COVID-19 respiratory transcriptomic datasets exist, none include adjudication of 2°BP status using a rigorous and standardized definition.

Taken together, our study sheds light on the microbial dynamics and host immune responses of 2°BP, a clinically important and serious complication of COVID-19 and other viral respiratory illnesses. Future studies are needed to validate these findings, clarify mechanisms in cohorts with other viral infections, and evaluate the diagnostic potential of metatranscriptomics for early detection of 2°BP.

## Methods

### Ethics

We studied patients enrolled in the COVID-19 Multiphenotyping for Effective Therapy (COMET) prospective cohort study of critically ill patients with acute respiratory illnesses at the University of California, San Francisco (UCSF) and Zuckerberg San Francisco General Hospital[12,22]. Patients were enrolled following research protocol #20-30497, which was approved by the UCSF Institutional Review Board (IRB).

With respect to enrollment and consent protocols in the COMET study, if a patient met inclusion criteria, then the study coordinator or physician obtained written or witnessed verbal informed consent for enrollment from the patient or their surrogate. Witnessed verbal consent was critical in the setting of COVID-19 when isolation procedures and limited personal protective equipment prevented in-person contact. Patients or their surrogates were provided with detailed written and verbal information about the goals of the study, the data and specimens that would be collected and the potential risks to the subject. Patients and their surrogates were also informed that there would be no benefit to them from being enrolled in the study and that they could withdraw informed consent at any time during the course of the study. All questions were answered and informed consent was documented, with a copy of the completed consent form provided to the patient or their surrogate.

Many critically ill patients are unconscious due to their underlying illness and/or are endotracheally intubated for airway management or acute respiratory failure. The patients who are not unconscious are often in pain and may have acute delirium due to critical illness and/or medications. As such, some patients are unable to comprehend, or it may be inappropriate to discuss the details of a complex research trial at the time of enrollment. For these reasons, many subjects are unable to provide informed consent at the time of enrollment. Because the study could not practically be done otherwise and was deemed to be minimal risk by the UCSF IRB, if a patient was unable and a surrogate was not available to provide consent, patients were enrolled with waiver of initial consent (explained below), including the collection of biological samples. UCSF's IRB/Committee on Human Research provided approval for enrolling these patients with waiver of initial consent.

Specifically, subjects who were unable to provide informed consent at the time of enrollment could have biological samples as well as clinical data from the medical record obtained prior to consent. Surrogate consent was vigorously pursued for all patients; moreover, each patient was regularly examined to determine if and when s/he would be able to consent for him/herself, and the nursing and ICU staff were contacted daily for information about surrogates' availability. For patients whose surrogates provided informed consent, follow-up consent from the patient was then obtained if they survived their acute illness and regained the ability to consent. For subjects who died prior to consent being obtained, a full waiver of consent was approved. Lack of a surrogate to provide consent is common in critically ill patients and may be particularly common among underserved populations, who would therefore be understudied were provisions not made to include these subjects. To address these concerns of generalizability and justice within the study sample, our IRB also approved a full waiver of consent for subjects in the study who remained unable to provide informed consent and had no contactable surrogate identified within 28 days. Before utilizing this waiver, we made and documented at least three separate attempts to identify and contact the patient or surrogate over a month-long period. Most patients enrolled in our study were consented by typical processes. No protected health information or personally identifiable information is provided as a part of this manuscript for any enrolled patients.

### Study design

We conducted a prospective case-control study of hospitalized adults requiring mechanical ventilation for COVID-19 with or without 2°BP (Fig. 1). All 397 patients with clinical polymerase chain reaction (PCR)-confirmed SARS-CoV-2 infection enrolled in the UCSF COMET observational study described above were initially considered for inclusion. Seventeen patients were co-enrolled in the National Institute of Allergy and Infectious Diseases-funded Immunophenotyping Assessment in a COVID-19 Cohort (IMPACC) Network study.

### Secondary bacterial pneumonia adjudication and patient inclusion

Amongst the 112 critically ill COVID-19 patients requiring mechanical ventilation enrolled in COMET, 44 cases of culture-confirmed 2°BP were adjudicated by 3 study team infectious disease physicians (NS, CD, CRL) based on the United States Centers for Disease Control and Prevention PNEU1 surveillance definition of pneumonia[31] and all available clinical data in the electronic medical record, blinded to metatranscriptomic results. Study team physicians also assessed the antimicrobial susceptibility patterns of cultured microbes based on phenotypic testing performed in the study site clinical microbiology laboratories, and identified days that patients with 2°BP had received antibiotics to which their cultured microbes were susceptible and/or resistant (Supplementary Data 1). Lab-reported susceptibilities as "susceptible dose dependent" or "intermediate" were included as "resistant" given the possibility of clinical treatment failure. *Enterobacteriaceae* known to harbor inducible *ampC*-class genes (e.g., *Enterobacter* species, *Klebsiella aerogenes*, *Citrobacter* species[56]), were considered resistant to 1st-3rd generation cephalosporins and piperacillin-tazobactam regardless of phenotypic susceptibility.

The date of 2°BP clinical diagnosis was set as the date on which the positive bacterial culture was ordered by the treating medical team. Patients with no clinical evidence of bacterial pneumonia at any point during their hospitalization (No-BP group, $N = 41$) were also identified. TA samples with metatranscriptomic data passing minimum quality control parameters (see below) were available for 29 2°BP patients and 29 No-BP patients. Within the 2°BP group, two additional patients were excluded from analyses as they did not have any TA samples collected within a 5-day window of 2°BP clinical diagnosis (-3 days to +2 days). The remaining final cohort of 2°BP ($N = 27$) and No-BP ($N = 29$) patients was leveraged for all metatranscriptomic analyses, which included all samples available within -7 or +7 days of clinical diagnosis.

### Tracheal aspirate and nasal swab sampling

Following enrollment, tracheal aspirate (TA) was collected periodically following intubation without addition of saline wash, and mixed 1:1 with DNA/RNA shield (Zymo Research, Cat. No R1100) in tubes containing bashing beads (Zymo Research, Cat. No S6012-50). Nasal swabs were collected into tubes prefilled with DNA/RNA shield. Samples were frozen within one hour and stored at −80 °C until nucleic acid extraction. Due to challenges with personnel availability in the setting of the COVID-19 pandemic, daily sampling was not possible for all patients.

### RNA sequencing

To evaluate host and microbial gene expression, metatranscriptomic RNA sequencing (RNA-seq) was performed on TA specimens. Following RNA extraction from 320 mL of input sample using the Zymo Pathogen Magbead Kit (Zymo Research, Cat. No R2146) and DNase treatment, human cytosolic and mitochondrial ribosomal RNA was depleted using FastSelect (Qiagen, Cat No. 334386). To control for background contamination, we included negative controls (water and HeLa cell RNA) as well as positive controls (spike-in RNA standards from the External RNA Controls Consortium (ERCC), Thermo Fisher Scientific, Cat. No 4456740) at 2.5 pg or 25 pg[57]. RNA was then

fragmented and underwent library preparation using the NEBNext Ultra II RNA-seq Kit (New England Biolabs, Cat. No E7770L). Libraries underwent 146 nucleotide paired-end Illumina sequencing on an Illumina Novaseq 6000. This generated an average of $7.99 \times 10^7$ total reads per sample (range: $0.51 \times 10^7$–$15.00 \times 10^7$), and an average of $3.11 \times 10^6$ bacterial reads per sample (range $1.19 \times 10^3$–$24.13 \times 10^6$.

## Quality control and mitigation of environmental contaminants

To minimize inaccurate taxonomic assignments due to environmental and reagent derived contaminants, non-templated "water only" and HeLa cell RNA controls were processed with each group of samples that underwent nucleic acid extraction. These were included, as well as positive control clinical samples, with each sequencing run. Negative control samples enabled estimation of the number of background reads expected for each taxon. Taxa mapping to the NCBI NT or NR databases at <0.1 rpM were excluded, as were microbes previously established as metagenomic contaminants (*Bradyrhizobium, Ralstonia, Delftia, Cutibacterium, Methylobacterium, Acidovorax, Chryseobacterium, Burkholderia, Sphingomonas*)[58,59].

A single sample per patient was utilized for analyses that involved direct comparison at a single timepoint, including bacterial RNA mass, alpha diversity, beta diversity, top pathogen identification, and TA only, host gene expression analyses. Samples were excluded from analysis if they had fewer than 1000 total reads mapping to bacterial taxa or a bacterial mass of <1 pg. The TA or NS samples collected closest to the date of clinical 2°BP diagnosis, defined as the date the positive culture was ordered by the clinical treatment team, were used in the primary timepoint analyses. To identify No-BP control samples appropriately matched by time from intubation, we selected samples collected closest to the median days post-intubation of the comparable 2°BP TA ($N = 6$ days) or NS ($N = 5$ days) samples. Because both host and microbial analyses were carried out for TA single timepoint comparison analyses, those samples were required to meet both host and microbe quality control standards.

## Statistics

Statistical significance was defined as a nominal $P < 0.05$ using two-tailed tests. Values less than $P < 0.0001$ were represented as <0.0001; all other P values are given with two significant digits. Categorical data were analyzed by Fisher's exact test and nonparametric continuous variables were analyzed by Mann–Whitney. Statistical approaches used for gene expression and microbiome analyses are detailed in each respective Methods section.

## Lung and nasal microbiome analyses

Taxonomic alignments were obtained from raw sequencing reads using the CZ ID pipeline (v8.3)[60,61], which performs quality filtration and removal of human reads followed by reference-based taxonomic alignment against sequences in the National Center for Biotechnology Information (NCBI) nucleotide (NT) database, followed by assembly of reads matching each taxon detected. Taxonomic alignments underwent background correction for environmental contaminants, and data was then aggregated to the genus level before calculating diversity metrics. SARS-CoV-2 viral load measured as *Betacoronavirus* rpM was determined from background-corrected metatranscriptomic taxon counts generated from the CZ ID pipeline.

We then calculated bacterial relative abundance (measured in rpM) and bacterial mass as described below in the "RNA mass calculations" section. We compared bacterial mass, alpha diversity, and beta diversity for tracheal aspirate samples from a single timepoint between patients with and without 2°BP. Alpha diversity was calculated from bacterial relative abundance based on SDI, and beta diversity was assessed based on the Bray-Curtis dissimilarity index and the Jaccard distance, using PERMANOVA to assess statistical significance, and using the R package *Vegan* (v.2.6.4)[62]. To account

for possible confounding, we performed four additional alpha diversity analyses and four additional beta diversity analyses; each analysis included one of the following covariates: (1) days of steroid receipt, (2) days of mechanical ventilation, (3) SARS-CoV-2 reads per million, (4) bacterial mass. We also compared beta diversity between nasal and tracheal aspirate samples from a single timepoint while accounting for intra-participant clustering, stratified by 2°BP status. Beta diversity results were plotted using non-metric multi-dimensional scaling (NDMS). Differential abundance analysis comparing tracheal aspirate samples from a single timepoint between patients with and without 2°BP was performed using the R package *DESeq2* (v1.36.0)[63] by assessing bacterial genera present in ≥30% of the samples. To account for receipt of steroids as a possible confounder to the differential abundance analysis, we performed a sensitivity analysis limited to patients who received steroids. To minimize the potential for false positive results with DESeq2[64], we required a Benjamini-Hochberg FDR < 0.001.

Similarly, we compared bacterial mass, alpha diversity, and beta diversity for nasal samples from a single timepoint between patients with and without 2°BP. We performed a Spearman's rho correlation analysis comparing the bacterial taxa paired nasal samples with tracheal aspirate samples for each patient. We described the median and IQR for the Spearman's rho correlation coefficients among patients with and without 2°BP.

Profiling of microbial metabolic pathways was carried out with HUMAnN (HMP Unified Metabolic Analysis Network) v3.9[65]. First, host-subtracted read 1 and read 2 fastq files were concatenated and used as input for the HUMAnN package. Next, the pathway abundances were normalized to relative abundance, and analyzed by the package MaAsLin 2.0 v1.16.0[66] (analysis_method = "LM", normalization = "NONE", transform = "LOG") to find metabolic pathways that are associated with 2°BP or No-BP patients. We only looked at unstratified pathways, and pathways expressed in at least 20% of the samples. Pathways with adjusted P-values below FDR < 0.25 were plotted in Supplementary Fig. 7.

## RNA mass calculations

Total bacterial RNA mass was calculated based on the ratio of total bacterial reads in each sample to total reads aligning to the ERCC RNA mass standards spiked into each sample[57] according to a previously established approach[67,68] employing the following equation: bacterial mass = [ERCC input mass × bacterial reads]/[ERCC reads]. Total RNA mass in each sample was calculated using the same approach but substituting total reads sequenced for total bacterial reads.

## AMR gene detection and genotype to phenotype correlations

Acquired ARGs annotated in the Antibiotic Resistance Gene-ANNOTation (ARG-ANNOT) database[69] were detected using the Short Read Sequence Typing (SRST2) algorithm[70]. *TEM-1D*, ubiquitously present in library preparation reagents and thus in negative control water samples, was excluded. We subsequently evaluated all AMR genes detected with ≥ 5% gene coverage or an average sequencing depth of ≥5 reads across the gene. We collapsed multi-mapped alleles for *mec, mcr, CTX-M, ampC and ampH* to the most abundant allele detected. Among patients with 2°BP, we described the longitudinal dynamics of pathogen-associated ARGs detected in tracheal aspirate samples collected 7 days before to 7 days after the 2°BP diagnosis. Gram-positive pathogen-associated ARGs included *BlaZ* and *mecA*; gram-negative pathogen-associated ARGs included *ACT-MIR, AmpC, CTX-M, FONA-1, MAL-CKO, OXA-1, OXA-50, OXY, RAHN-1, SED-1, Qnr-S, Mcr1* and *Mcr3*.

For genotype to phenotype analyses, we evaluated the correlation between detection of *mecA* and methicillin/nafcillin resistance in patients with *S. aureus* 2°BP (Supplementary Table 3). In addition, we evaluated the correlation between detection of *CTX-M* and ceftriaxone

resistance in patients with 2°BP due to *Enterobacteriaceae* (Supplementary Table 4).

## Host gene differential expression

Following demultiplexing, sequencing reads were pseudo-aligned with kallisto[71] to an index consisting of all transcripts associated with human protein coding genes (ENSEMBL release 99), long non-coding RNA, cytosolic and mitochondrial ribosomal RNA sequences, and the sequences of ERCC RNA standards. Gene-level counts were generated from the transcript-level abundance estimates using the R package tximport[72], with the scaledTPM method. For quality control, we only retained samples with a total of at least 1,000,000 estimated protein-coding gene counts, and a proportion of ribosomal RNA to total RNA ≤ 50%, according to a previously described approach[39]. In addition, we only analyzed host genes with at least 10 counts in at least 20% of samples.

Differential expression analysis was performed in R (v4.3.2) using the package limma-voom[73] (v3.58.1). For the comparison between 2°BP and No-BP patients, we adjusted for SARS-CoV-2 viral load rpM by adding the coefficient log10(rpM+1) to the linear model (without adjusting for any other covariates). For the analysis of host gene expression and the number of days on corticosteroid, we modeled gene expression on the number of days the patients had been receiving corticosteroid prior to sample collection (without adjusting for any other covariates). For the analyses of host gene expression and bacterial mass, we modeled gene expression on log10-transformed bacterial mass (without adjusting for any other covariates). For the analyses of corticosteroid recipients (or non-corticosteroid recipients), we restricted to patients who had (or had not) received treatment with steroids at any time prior to sample collection. Significant genes were identified using an FDR < 0.1. Differential expression analysis results are provided in Supplementary Data files as indicated.

GSEA was performed using the package fgsea (v1.28.0), and the Hallmark pathways were obtained from the package msigdbr (v7.5.1). The t-statistics (obtained from limma's differential expression analysis) were used to rank all genes, and used as input for the fgseaMultilevel function (minSize = 15, maxSize = 500). Pathways with adjusted *P* value below 0.05 were considered statistically significant.

## Reporting summary

Further information on research design is available in the Nature Portfolio Reporting Summary linked to this article.

# Data availability

Source data are provided with this paper. The raw fastq files with microbial sequencing reads are available under NCBI BioProject ID: PRJNA1033689. The host gene counts are available under NCBI Gene Expression Omnibus (GEO) accession number: GSE246795. The human raw sequencing data are protected due to data privacy restrictions from the IRB protocol governing patient enrollment, which protects the release of raw genetic sequencing data from those patients enrolled under a waiver of consent. To honor this, researchers who wish to obtain raw fastq files for the purposes of independently generating gene counts can contact the corresponding author (chaz.langelier@ucsf.edu) and request to be added to the IRB protocol. All patient demographic data, sample metadata, processed microbial sequencing reads, and processed host gene counts needed to replicate this study are available in the code inputs folder, and all data generated in this study are available in the code outputs file and in the Source Data file. Source data are provided with this paper.

# Code availability

All code and source data used for analyses can be found at https://zenodo.org/records/13786733. Source data are also provided with this paper.

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

## Acknowledgements

This project was funded in part by the National Institutes of Health (U19AI077439, supporting the UCSF component of the NIAID Immunophenotyping Assessment in a COVID-19 Cohort [IMPACC] Network). We would like to thank the full COMET and IMPACC network consortia for their support and feedback in this work. This work was additionally supported in part by the Chan Zuckerberg Biohub (CRL, JLD) and the following grants from the NHLBI: R01HL155418 (C.R.L., P.M.M.), R35HL140026 (C.S.C., L.P.N.), F32HL151117 and K23HL163491 (A.S.).

## Author contributions

N.S. and A.T. contributed equally. N.S., A.T., and C.R.L. conceived of, designed, led, executed the study and wrote the manuscript. N.S., A.T., V.T.C., H.V.P., C.D., and C.R.L. performed the data analysis. C.L., R.G., A.D., and N.N. generated the metatranscriptomic data. S.C.H. provided support in data and study management. C.S.C., D.J.E., M.F.K., C.M.H., P.G.W., and C.R.L. led the COMET study. V.T.C., H.V.P., C.S.C., D.J.E., M.F.K., C.M.H., P.G.W., M.A.M., J.B., B.S.Z., C.P.M., A.G., A.S., P.M.M., K.L.K., K.N.K., and J.L.D. provided advice on the analysis and edits to the manuscript. The COMET Consortium performed the COMET study including patient recruitment, sample collection, data generation, and data management. All authors were involved in the reviewing and editing of the manuscript.

## Competing interests

The authors declare no competing interests.

## Additional information

[1]Division of Infectious Diseases, Department of Medicine, University of California, San Francisco, CA, USA. [2]Department of Pediatrics, University of California, San Francisco, CA, USA. [3]Chan Zuckerberg Biohub San Francisco, San Francisco, CA, USA. [4]Division of Pulmonary, Department of Medicine, Critical Care, Allergy and Sleep Medicine, University of California, San Francisco, CA, USA. [5]Department of Medicine, University of Washington, Seattle, WA, USA. [6]University of Texas, Austin, TX, USA. [7]Department of Pediatrics, Arkansas Children's, Little Rock, AR, USA. [8]Chan Zuckerberg Initiative, Redwood City, CA, USA. [9]Department of Biochemistry and Biophysics, University of California, San Francisco, CA, USA. [10]UCSF CoLabs, University of California, San Francisco, CA, USA. [11]Lung Biology Center, University of California, San Francisco, CA, USA. [12]Department of Medicine, University of California, San Francisco, CA, USA. [13]Department of Pathology, University of California, San Francisco, CA, USA. [38]These authors contributed equally: Natasha Spottiswoode, Alexandra Tsitsiklis. ✉e-mail: chaz.langelier@ucsf.edu

# COMET Consortium

Saharai Caldera[1], Catherine DeVoe [1], Sarah B. Doernberg[1], Rajani Ghale[1,4], Christina Love[1], Eran Mick[1,3], Charles R. Langelier [1,3] ✉, Hoang Van Phan[1], Paula Hayakawa Serpa[1], Natasha Spottiswoode [1,38], Alexandra Tsitsiklis[1,38], Deanna Lee[4,14], Maira Phelps[3], Carolyn S. Calfee[4,12,15], Suzanna Chak[4,16], Stephanie Christenson[4], Walter L. Eckalbar[4,10,12], David J. Erle[4,10,11,12], Sidney C. Haller[4], Carolyn M. Hendrickson [4], Alejandra Jauregui[4], Chayse Jones[4], Carolyn Leroux[4], Michael Matthay[4,12,14,15], Lucile P. A. Neyton[4], Viet Nguyen[4,14], Austin Sigman[4], Aartik Sarma [4], Andrew Willmore[4], Prescott G. Woodruff[4], Beth Shoshana Zha [4], Michael Adkisson[10], Saurabh Asthana[10,13,17], Zachary Collins[10,13,17], Gabriela K. Fragiadakis[10,18], Lenka Maliskova[10], Ravi Patel[10], Arjun Rao[10,13,17], Bushra Samad[10,13,17], Andrew Schroeder[10], Cole Shaw[10,17], Kirsten N. Kangelaris[12,16], Matthew F. Krummel [13], Divya Kushnoor[13], Tasha Lea[13], Kenneth Hu[13], Alan Shen[13,17,19,20], Jessica Tsui[13,17,19,20], Raymund Bueno[18], David Lee[18], Yang Sun[18], Erden Tumurbaatar[18], Alyssa Ward[18], Monique van der Wijst[18,21,22], Jimmie Ye[18,21,23,24], K. Mark Ansel[25,26], Vincent Chan[25], Kamir Hiam[25,27,28,29], Elizabeth McCarthy[25,27], Priscila Muñoz-Sandoval[25,26], Anton Ogorodnikov[25], Matthew Spitzer[25,28], Wandi S. Zhu[25,26], Gracie Gordon[21,30,31], George Hartoularos[31], Sadeed Rashid[27], Nicklaus Rodriguez[27], Kevin Tang[27], Luz Torres Altamirano[27], Alexander Whatley[32], Yun S. Song[32,33], Aleksandra Leligdowicz[16], Michael Wilson[34], Nayvin Chew[17,19,20], Alexis Combes[17,19,20], Tristan Courau[17,19,20], Norman Jones[35], Jeff Milush[35], Nitasha Kumar[35], Billy Huang[36], Salman Mahboob[36], Randy Parada[36] & Gabriella Reeder[17,20,37]

[14]Cardiovascular Research Institute, University of California, San Francisco, CA, USA. [15]Department of Anesthesia, University of California, San Francisco, CA, USA. [16]Department of Medicine, University of Toronto, Toronto, ON, Canada. [17]Bakar ImmunoX Initiative, University of California, San Francisco, CA, USA. [18]Division of Rheumatology, University of California, San Francisco, CA, USA. [19]Department of Anatomy, University of California, San Francisco, CA, USA. [20]Disease 2 Biology CoLab, University of California, San Francisco, CA, USA. [21]Institute for Human Genetics, University of California, San Francisco, CA, USA. [22]Department of Genetics, University of Groningen, University Medical Center Groningen, Groningen, The Netherlands. [23]Department of Epidemiology and Biostatistics, University of California, San Francisco, CA, USA. [24]Institute of Computational Health Sciences, University of California, San Francisco, CA, USA. [25]Department of Microbiology and Immunology, University of California, San Francisco, CA, USA. [26]Sandler Asthma Basic Research Center, University of California, San Francisco, CA, USA. [27]Helen Diller Family Comprehensive Cancer Center, University of California, San Francisco, CA, USA. [28]Department of Otolaryngology, University of California, San Francisco, CA, USA. [29]Parker Institute for Cancer Immunotherapy, San Francisco, CA, USA. [30]Department of Bioengineering and Therapeutic Sciences, University of California, San Francisco, CA, USA. [31]Biological and Medical Informatics Graduate Program, University of California, San Francisco, CA, USA. [32]Department of Electrical Engineering and Computer Sciences, University of California, Berkeley, CA, USA. [33]Department of Statistics, University of California, Berkeley, CA, USA. [34]Department of Neurology, Weill Institute for Neurosciences, University of California, San Francisco, CA, USA. [35]Division of Experimental Medicine, Core Immunology Laboratory, University of California, San Francisco, CA, USA. [36]Department of Orofacial Sciences, School of Dentistry, University of California, San Francisco, CA, USA. [37]Biomedical Sciences Graduate Program, University of California, San Francisco, CA, USA.

