## [Transparent Peer Review file · Nature Communications]

Microbial Dynamics and Pulmonary Immune Responses in COVID-19 Secondary Bacterial Pneumonia

Corresponding Author: Dr Charles Langelier

Version 0:

Reviewer comments:

Reviewer #1

(Remarks to the Author)

The manuscript by Spottiswoode et al. provides a metatranscriptomic analysis of tracheal and nasal swab samples from a prospective cohort of 56 critically ill adults intubated for COVID-19 whose samples were available for analysis, stratified by clinical diagnosis of secondary bacterial pneumonia. While I commend the authors for undertaking a complex study of a sizeable cohort of critically ill subjects, and correlating host and pathogen signals, there are several major concerns with the manuscript as currently written. Specifically, the diagnosis of 2BP is confounded by steroid use. It is unclear exactly how or if the authors accounted for this in their analysis since there were only 6 subjects who did not receive steroids and all of them did not develop 2BP. Additional significant concerns are outlined below.

Major concerns:

1. How did the study team decide when or how often subjects would be sampled (or was this at the discretion of the clinical teams caring for the patients)? How does this variation in sampling affect the outcome of the analyses?
2. There are limitations to studying rank abundance longitudinally, since this is influenced by the amount of RNA detected at one time point vs. another, as well as the abundance of the other bacteria in each sample. This is a particular concern when an unknown number of these other bacteria are susceptible to one or more antibiotics given to the patient over the various time points. This limitation is likely a major driver of the inconsistent results observed in Figure 3.
3. NS and TA co-detection of pathogens is under-analyzed. Is organism concordance between the two anatomic sites more likely when there is a 2BP vs. no 2BP? Does the microbiome (not only the culture-confirmed pathogen) concordance between the two anatomic sites depend on the presence of pneumonia, other clinical factors? Absent a more thorough comparison of these two compartments I don't think we can conclude that "dominance of pathogenic bacteria can be co-detected in the nares" (lines 307-308).
4. Use of steroids confounds the results. There is no analysis of the impact of steroids in those who develop 2BP vs those who do not develop 2BP. It is difficult to discern which of these transcriptomic changes is related to steroid use in a background of 2BP, since 97.7% of patients with 2BP received corticosteroids (0% in the transcriptomic cohort). The conclusion that the 2BP lower airway transcriptome is characterized by suppressed TNF α signaling is not well-supported by the data, since suppressed TNF α signaling is not seen in those who received steroids (unclear who this control group is—the 2.3% or 0% of subjects with 2BP who didn't receive steroids?). Furthermore, this discrepancy based on steroid use is not further explored.
5. The microbiome is not fully analyzed here—just alpha diversity and one NMDS plot with a non-significant p-value. The conclusion [lines 332-333] that "2BP further disrupts microbial community structure in patients with existing SARS-CoV-2 infection" is not supported by the NMDS plot nor is it fully explored.
6. There is a small literature on microbiome concordance between the lower airways and nasal swabs. It would be helpful to briefly discuss this within the context of your metatranscriptomic results. Notably, there is much more concordance between the oral microbiome (another non-invasive surrogate site) and the lower airways than there appears to be with the nasal microbiome.
7. Lines 364-367: "Our data suggest that screening for other clinically important AMR genes may prove useful for identifying patients with drug-resistant pneumonia, with implications for both antimicrobial stewardship and infection control." This is intriguing and may have significant clinical implications in the future. However, I would urge caution about over-interpretation of these results. In what percent of patients were the AMR genes identified in the NS samples of clinical relevance (i.e, found in both the lower airways and also in the organism causing the pneumonia)? What are the positive and negative predictive values of AMR genes found in TA samples? For instance, it would be unfortunate if a patient received a very broad antibiotic

as a result of the NS AMR findings, when in fact their clinical pneumonia was due to different organism which did not harbor the AMR gene. The example of *mecA* nares screening is more nuanced than is suggested here in the discussion. Screening for MRSA is done in the nares because 1) the nares are a recognized reservoir of *S. aureus* (whereas many gram negative causes of pneumonia are not found in the nose) and 2) *mecA* screening is used for its negative predictive value, and not its positive predictive value.

8. A significant limitation of the manuscript is the difficulty understanding which comparisons are being made and discussed, especially when steroid use confounds 2BP diagnosis. For instance, when evaluating the effects of steroid use, is the analysis limited to only those with 2BP? Or are all samples included regardless pneumonia diagnosis? The comparisons made in figure 6c seem to exclude individuals who did not receive steroids, however that doesn't account for the fact that steroids use is significantly confounded by 2BP diagnosis (and vice versa).

9. Tables S2-S8 are missing from the supplementary materials file. I have questions about the data contained in them (specifically Tables S5 and S6 as described above in #4 and #8). Without these tables or additional info in the main text (described above) I'm unable to thoroughly review the manuscript.

Minor concerns:

10. Line 64: pathogens and the lung microbiome are both mentioned as though the two are separate entities. Do the authors wish to argue that pathogens can be analyzed separately from the lung microbiome? This sentence also asserts that there is a dynamic relationship between microbiome and immune response, but that this dynamic relationship has not been studied in much depth. I suggest re-phrasing this to acknowledge that one cannot assert a relationship is dynamic unless it has been adequately studied.

11. Tracheal aspirate (TA) should be defined at the first use in the manuscript.

12. Line 104: "increased bacterial RNA load"—does this refer to rRNA gene sequences or RNA transcripts from bacteria?

13. Figure 1b: please clarify if these culture results represent only tracheal aspirates, or if nasal swab cultures are represented here as well. If nasal swabs are represented here, how did the study team adjudicate discordant culture results (TA vs. NS)?

14. Please edit to exclude scientific notation for p-values in the manuscript text (e.g., $P = 3.03e-5$).

15. In Figure 3, patient 1114 does not illustrate the stated finding. *Pseudomonas* is the #1 ranked pathogen in all samples shown here, so it cannot be described as "expanding". Additionally, it is overly simplistic to say that in patient 1231 initiation of antibiotics resulted in falling pathogen abundance, since the pattern of falling *Escherichia* abundance was already established before susceptible antibiotics were started.

16. Line 214, please correct the p value given here.

17. Figure 4D. My version of the manuscript has random icons in place of the x- and y-axis labels.

18. Figure 3 and 5, patient 1154 with *K. aerogenes* and an inducible *ampC* beta lactamase treated with pip-tazo: I'd argue that this individual did not receive an antibiotic to which his or her organism was susceptible. (See question 2.2 in <https://www.idsociety.org/practice-guideline/amr-guidance/#AmpC%CE%B2-Lactamase-ProducingEnterobacterales>).

19. I would consider changing the patient numbering system in the manuscript. Patient 1145 and 1154 both have *Klebsiella* 2BPs and are heavily discussed here. The use of such similar numbers in the figures and text may lead to unnecessary confusion.

20. Line 278: The exact number of differentially expressed genes is missing.

21. Lines 279-280: I'm unclear how NOT finding suppressed TNF α in those with 2BP and steroid use supports the hypothesis that corticosteroid treatment might contribute to the suppressed TNF α signaling in 2BP.

22. I highly commend the authors for using such a rigorous definition of pneumonia.

23. Discussion [lines 320-329]: While I agree that metatranscriptomics has the potential to enable early detection of 2BP, I do not think the data presented here provide strong evidence of this potential. Please temper expectations in this paragraph to ensure that readers understand the limited predictive capability of the NS transcriptomics data presented here.

Reviewer #2

(Remarks to the Author)

In this manuscript, the authors evaluate the upper and lower microbiome of critical COVID-19 patients undergoing mechanical ventilation through longitudinal sampling. The patients in this study are divided in two groups: those having secondary bacterial pneumonia (2BP), as determined by clinical evaluation and microbial culture, and those not presenting it. They evaluate the metatranscriptomics potential in detecting secondary pathogens in patients with 2BP, and determine the dynamics of these pathogens, as well as potential interactions between the microbiota and the host in these samples. One of their major claims is that there is a link between corticosteroid treatments, host gene expression pathways, and bacterial mass in the respiratory airways, which could lead to 2BP.

The manuscript presents promising results, but as a reviewer there are some issues that I would like to raise before article acceptance.

1- Given that metatranscriptomics analysis obtains the information from active bacterial/host transcripts, providing an analysis of bacterial functions in the samples analyzed, and whether they differ in 2BP and no-BP or with other covariates would be of utmost interest for the readers in the manuscript. Especially since the microbial taxonomic composition does not differ significantly between 2BP and no-BP, knowing whether or not there are significant functional shifts can bring novel insights into the understanding of 2BP in these patients.

2- In line 177, when discussing the compositional differences between two groups based on a PERMANOVA analyses, the wording "approached statistical significance" is not appropriate. If the chosen cut-off to determine statistical significance is set to 0.05 as indicated in the methods, in this case the differences are not statistically significant and the wording in the results should not mislead the reader. In addition, the discussion indicates that one of the findings of the manuscript is that 2BP is characterized by a disruption of the lung microbiome, which is not shown by this analysis. Therefore, the claims from

the manuscript should be modified accordingly.

- 3- The code provided in the manuscript, available in a Github repository, is incomplete. In the README file, 3 *.R files are indicated to reproduce the analyses in the manuscript, but two of these files are missing, presumably removed in an update of the repository. Therefore, it is not possible to assess the analyses and the computational methods performed.
- 4- Related to the issue above, the detection of AMR genes is not described in the methods. Which specific databases or functional annotations are used to identify these genes in the samples? Also, determination of SARS-CoV-2 viral load is not reported.
- 5- More details in the microbiome analysis should be reported. Besides aggregation at the genus level, is there any normalization or transformation applied before the different analyses performed? Additionally and most importantly, no association between the microbial composition and potential confounding factors has been reported. Do the antibiotics, SARS-CoV-2 viral load, bacterial load, corticosteroid treatment or other factors influence the taxonomic composition in the samples evaluated? It is intriguing, since in the host differential expression analyses (2BP vs no-BP) viral load is included as a covariate. Additionally, the discussion states that patients developing 2BP were more likely to have received corticosteroids. A thorough analysis of potential confounders or variables affecting the composition (and the function, see comment 1) should be performed.
- 6- A more thorough investigation between the interaction of bacterial mass, corticosteroids and host gene expression should be performed. In the case of the association of host gene expression with bacterial mass, this is only performed in 2BP patients. Is it possible to deconfound bacterial mass and corticosteroid treatment? Also, what is the association of host gene expression with bacterial mass in no-BP patients?
- 7- In the discussion, it is stated that this study includes the novel use of host-microbe metatranscriptomics to study secondary bacterial infections, but other previous works have also employed metatranscriptomics to identify potential secondary infections by oral bacteria (Sulaiman et al, Nat Microbiol 2021 as reported in the references of the manuscript; or Wang et al, Signal Transduction and Targeted Therapy 2023), or even single-cell transcriptomics data to identify bacterial pathogens and their associations with host cells (Llorens-Rico et al, Nat Communications 2021). Therefore, their novelty claim should be modified accordingly.
- 8- Bacterial RNA load/mass calculations are not clear, since the ratio reads/mass ratio can be affected by factors such as cell viability and transcriptional activity. Additionally, the amount of ERCC spiked into each sample should be detailed.
- 9- In figure 2G of the pdf, the color legend for the number of cases is blank, missing the information to interpret the plot.
- 10- In figure 2E, patient 1154 shows the bacterial abundance for E. coli, but in figure 3, this patient is indicated to have a 2BP caused by Klebsiella, which is the correct organism?
- 11- In line 228: "Among the 12 patients who had NS samples collected prior to the date of clinical 2^oBP diagnosis, 8/9 (88.9%)", is it 12 patients or 9?
- 12- Figure 4D has corrupt x- and y- axis labels as well as panel titles and bottom-right legend.
- 13- Supplementary figures and tables numbering should be revised. In line 259 the authors refer to Figure S2 but I assume it is figure S3? Also in line 415 the authors point to Table S5 but it should probably refer to table S2?
- 14- In line 278 the number of genes differentially expressed is missing ("4XX"). According to the table S5 it should be 421?
- 15- The average (and range) of reads sequenced per library (total reads as well as reads assigned to bacteria) should be reported, for future users to be able to produce comparable results in similar samples.
- 16- I noticed that references 51 and 54 are the same (duplicated), so a revision of the references is needed

Reviewer #3

(Remarks to the Author)

This article delves into the metagenomic transcriptomic analysis of tracheal aspirate samples from COVID-19 mechanically ventilated patients afflicted with secondary bacterial pneumonia (2BP).

It also offers insights from partial metagenomic transcriptomes of nasopharyngeal swabs. Furthermore, by integrating transcriptomic data from human samples obtained from tracheal aspirates, likely epithelial cells, it provides a nuanced exploration of the microbial dynamics and immune features in COVID-19 patients experiencing secondary bacterial pneumonia.

While the clinical data within the manuscript are meticulously detailed and professionally elucidated, the depth of data analysis and exploration appears somewhat restrained. Nonetheless, the longitudinal cohort comprising multiple specimens enriches the study's scientific value. Notably, the absence of comprehensive microbiome characterization regarding 2BP underscores the significance of this study in broadening our understanding of this condition.

Here are some refined suggestions and inquiries:

1. Could there be a notable contrast in the rate of decline between 2BP and non-2BP cases in Figure 2D, despite the observed reduction in diversity in both scenarios?
2. The classification of the small panels depicting individual patient statuses in Figure 3 into categories such as "pathogen expansion," "pathogen resistance," "decrease in pathogen abundance," and "most persistent" would enhance clarity. Representative samples from each category could be highlighted, with additional data relegated to the appendix.
3. Although the co-occurrence of secondary bacterial pneumonia with COVID-19 remains understudied, are there discernible parallels between the findings of this analysis and existing research on COVID-19 co-infections?
4. Given the meticulous documentation of antibiotic usage and resistance genes, could an integrated analysis further elucidate potential correlations between identified resistance genes and clinical outcomes? For instance, were any distinctive resistance gene patterns identified in cases involving *Pseudomonas aeruginosa* infection, as discussed?
5. In the human transcriptomic analysis section of the results, the tacit assumption regarding the potential contribution of

corticosteroid therapy lacks explicit clarification. Moreover, the sudden exploration of the "association between bacterial RNA load and host gene expression" without prior groundwork warrants elucidation. Additionally, while the method part delineates bacterial load calculation, specifying which samples were classified as high versus low load would enhance clarity.

Some minor issues warranting attention include:

1. Potential overlay of different images in Figure 4C.
2. Ambiguity surrounding the mention of "4XX genes" in line 278.

Version 1:

Reviewer comments:

Reviewer #1

(Remarks to the Author)

The revised manuscript by Spottiswoode et al. provides a metatranscriptomic analysis of tracheal and nasal swab samples from a prospective cohort of 56 critically ill adults intubated for COVID-19, stratified by clinical diagnosis of secondary bacterial pneumonia. I commend the authors for undertaking a complex study of a sizeable cohort of critically ill subjects, and correlating host and pathogen signals. The current revision is a significant improvement and substantially addresses my previous concerns. The new microbiome analyses offer a more complete picture of the data and the analysis of steroid use (a potential confounding factor in 2BP diagnoses) addresses many of my concerns. I have no significant outstanding concerns.

Reviewer #3

(Remarks to the Author)

The authors provided thorough responses to my five main questions and addressed minor issues accordingly. Their detailed feedback subtly underscored the data's reproducibility. This study examines microbiome and host transcriptome data, distinguished by its rigorous experimental design including longitudinal sample cohorts and comprehensive clinical data. While focused on COVID-19 patients, its methodology holds relevance for understanding diseases caused by microorganisms. Furthermore, the study offers empirical validation through NGS data, despite longstanding clinical speculations regarding the impact of immunosuppressants on opportunistic pathogens.

REVIEWER COMMENTS

Reviewer #1 (Remarks to the Author):

The manuscript by Spottiswoode et al. provides a metatranscriptomic analysis of tracheal and nasal swab samples from a prospective cohort of 56 critically ill adults intubated for COVID-19 whose samples were available for analysis, stratified by clinical diagnosis of secondary bacterial pneumonia. While I commend the authors for undertaking a complex study of a sizeable cohort of critically ill subjects, and correlating host and pathogen signals, there are several major concerns with the manuscript as currently written. Specifically, the diagnosis of 2BP is confounded by steroid use. It is unclear exactly how or if the authors accounted for this in their analysis since there were only 6 subjects who did not receive steroids and all of them did not develop 2BP. Additional significant concerns are outlined below.

We appreciate R1's input and feedback, which we have directly incorporated. Most notably, we have taken several steps to evaluate the impact of steroid treatment and other potential confounders on our findings.

With respect to steroid treatment, we would like to clarify that 12 subjects did not receive steroids (2/27 patients in the 2°BP group and 10/29 patients in the No-BP group) prior to the date of sample collection in our primary comparative analyses of 2°BP patients versus time-matched samples from No-BP patients (Supp. Table 1b).

We discuss below specific additional analyses that we have carried out to address the impact of steroid use on our results.

Major concerns:

1. How did the study team decide when or how often subjects would be sampled (or was this at the discretion of the clinical teams caring for the patients)?

We appreciate the chance to clarify this point. The intent in this study was to collect samples daily. However, due to a variety of pandemic-related constraints, including short staffing due to illness, limited staff availability during weekends, and other factors, samples were collected less frequently than daily. Despite this, our cohort represents one of the largest cohorts of intubated COVID-19 patients and one of the few with rigorously adjudicated secondary bacterial infections. We have added the following text to better clarify:

Line 525: "Due to challenges with personnel availability in the setting of the COVID-19 pandemic, daily sampling was not possible for all patients."

How does this variation in sampling affect the outcome of the analyses?

Variations in sample collection reduced the number of 2°BP patients with analyzable samples near the date of 2°BP onset, and the number of analyzable longitudinal samples. This likely limited our ability to detect microbiome and host transcriptional differences that may have existed between 2°BP patients and controls. Variations in sampling also limited the resolution of our assessments of pathogen longitudinal dynamics. We have now highlighted these points as limitations in the discussion as follows:

Line 471: "Limitations include a relatively small sample size and incomplete longitudinal sampling for all patients, which reduced the number of patients with analyzable samples

near the date of 2°BP onset, and the number of analyzable longitudinal samples. This likely limited our ability to detect microbiome and host transcriptional differences that may have existed between 2°BP patients and controls.”

2. There are limitations to studying rank abundance longitudinally, since this is influenced by the amount of RNA detected at one time point vs. another, as well as the abundance of the other bacteria in each sample. This is a particular concern when an unknown number of these other bacteria are susceptible to one or more antibiotics given to the patient over the various time points. This limitation is likely a major driver of the inconsistent results observed in Figure 3.

We appreciate this feedback which have made a concerted effort to incorporate. Specifically, for our longitudinal analyses, we now evaluate pathogen mass normalized by total sample RNA mass, as well as pathogen mass alone, to directly address concerns about the influence of the amount of RNA detected at one sampling time point vs another, as well as the abundance of other bacteria in the sample. Given that rank of the culture-confirmed pathogen may still be of interest as it has relevance to clinical applications of mNGS (for which common algorithms are designed to detect the most abundant organism/pathogen in mNGS data), we still include rank for comparison.

Given the additional request from R3 to reduce the complexity of Figure 3 by highlighting only exemplary cases of distinct pathogen trajectories, Figure 3 now includes each of these three metrics (rank, pathogen mass, normalized pathogen mass) for the following pathogen trajectory scenarios:

- a. pathogen expansion in the lower airway prior to 2°BP diagnosis in the absence of any antibiotic treatment
- b. pathogen abundance reduction following 2°BP diagnosis
- c. pathogen persistence as the most abundant microbe in the lower airway despite initiation of appropriate antimicrobial therapy
- d. pathogen persistence as the most abundant microbe in the lower airway following treatment with antibiotics to which the pathogen was resistant

Complete longitudinal data for each patient with 1) normalized pathogen mass, 2) pathogen mass and 3) rank are now found as Supplementary Figures 8-10 for tracheal aspirate samples and Supplementary Figures 12-14 for nasal swab and tracheal aspirate samples.

Figure 3. Dynamics of 2°BP pathogen over time relative to the date of clinical diagnosis highlighting examples of unique pathogen trajectories. The top row includes plots of genus level 2°BP pathogen rank based on bacterial reads per million (rpM) in the lung microbiome for each of four pathogen trajectories (expansion, reduction, persistence and resistance). The middle row includes plots of pathogen mass. The bottom row consists of plots of pathogen mass normalized to the total RNA mass of sample. Days relative to 2°BP clinical diagnosis are plotted on the X axis. Days during which patient received antibiotics to which 2°BP pathogen was phenotypically susceptible (light grey bar) or resistant (dark grey bar) are plotted below. The patient ID and pathogen genus are listed above each plot. Open circles denote samples in which the pathogen was not detected by metatranscriptomics.

3. NS and TA co-detection of pathogens is under-analyzed. Is organism concordance between the two anatomic sites more likely when there is a 2BP vs. no 2BP? Does the microbiome (not only the culture-confirmed pathogen) concordance between the two anatomic sites depend on the presence of pneumonia, other clinical factors? Absent a more thorough comparison of these two compartments I don't think we can conclude that "dominance of pathogenic bacteria can be co-detected in the nares" (lines 307-308).

We appreciate the reviewer bringing up this point which we have addressed with two new analyses. The first assesses concordance between the upper and lower airway microbiome by Spearman's correlation test in 2°BP and No-BP patients:

D
Figure 4d. Stacked bar plots highlighting the taxonomic composition of paired NS and tracheal aspirate (TA) samples from 2°BP patients (top) and No-BP patients (bottom). Bacterial taxa present at <0.5% relative abundance across all samples were included in the “other” category, except if the microbe had been cultured as a 2°BP pathogen.

This analysis did not demonstrate a significant difference in upper versus lower airway microbiome concordance based on 2°BP status, consistent with a recent metatranscriptomics study of COVID-19 patients (Sulaiman et al. *Nature Microbiology*. 2021).

In addition, we compared Bray-Curtis indices between NS and TA samples in 2°BP and No-BP patients, respectively. This analysis yielded consistent results with the one described above, and did not show significant differences in beta diversity between the upper versus lower airway microbiome in either 2°BP or No-BP patients.

Supplementary Figure 15. Beta diversity comparing nasal swab to tracheal aspirate samples. Non-metric multidimensional scaling (NMDS) plots demonstrating upper versus lower airway beta diversity based on Bray-Curtis dissimilarity index between nasal swabs (NS, purple) and tracheal aspirate (TA, red), accounting for intra-participant clustering. **a** 2°BP patients and **b** No-BP patients. P values based on the PERMANOVA test with 1000 permutations.

In summary, while in several cases the culture-confirmed 2°BP pathogen was the most abundant microbe in both the upper and lower respiratory tract microbiome, strong concordance did not exist on the broader microbiome scale. We have thus revised the text to discuss this as follows:

Line 247: “We noted that in 14/15 (93.3%) of cases, the 2°BP pathogen was detected in at least one NS sample within 7 days of clinical diagnosis, and in 7/15 (46.7%) cases it ranked in the top 3 most abundant (Fig. 4c). Among the 9 patients who had NS samples collected prior to the date of clinical 2°BP diagnosis, 8/9 (88.9%) had at least one etiologic pathogen detected (Supp. Figs. 12-14). We observed a low to moderate concordance between upper and lower respiratory tract microbiomes that did not significantly differ based on 2°BP status (median Spearman’s rho 0.34, interquartile range (IQR) 0.13-0.54) for 2°BP patients versus 0.48 (IQR 0.26-0.61) for No-BP patients. Assessment of NS versus TA beta diversity based on Bray Curtis dissimilarity index also did not reveal significant differences based on sampling site in 2°BP or No-BP patients (Supp. Fig. 15).”

We have also revised the sentence highlighted by the reviewer as follows:

Line 385: “Using comparative metatranscriptomics, we found that 2°BP is characterized by changes in the lung microbiome and host transcriptome, as well as dominance of pathogenic bacteria in the lungs that in many cases could be co-detected in the upper airway.”

4. Use of steroids confounds the results. There is no analysis of the impact of steroids in those who develop 2BP vs those who do not develop 2BP.

It is difficult to discern which of these transcriptomic changes is related to steroid use in a background of 2BP, since 97.7% of patients with 2BP received corticosteroids (0% in the transcriptomic cohort). The conclusion that the 2BP lower airway transcriptome is characterized by suppressed TNFa signaling is not well-supported by the data, since suppressed TNFa signaling is not seen in those who received steroids (unclear who this control group is—the 2.3% or 0% of subjects with 2BP who didn't receive steroids?). Furthermore, this discrepancy based on steroid use is not further explored.

We appreciate this point and have taken several steps, including new analyses, to assess the impact of steroids on our results, as detailed below. We would also like to note that we did initially attempt to evaluate the impact of steroids on our host gene expression analysis by performing the following:

- 1) Differential gene expression and gene set enrichment analysis (GSEA) between all 2°BP and No-BP patients, regardless of steroid status (Figures 6a-6b)
- 2) A secondary differential gene expression analysis and GSEA between 2°BP and No-BP patients, restricted to only those patients who had received steroids prior to the time of sample collection (this reduced the comparison sample size by 2 2°BP patients and 10 No-BP patients) (Fig. 6c).

In the 'steroid-only' analysis, (#2 above) we attempted to assess differences in host response between 2°BP and No-BP patients when differential steroid treatment was removed as a variable. This was done because 25/27 2°BP patients received steroids prior to sampling while only 19/29 No-BP patients had received steroids prior to the matched sampling timepoint. One interpretation of our results (presented in Fig. 6c) is that without the imbalance in steroid treatment favoring 2°BP patients, TNFa signaling is no longer downregulated in 2°BP versus No-BP, suggesting but not proving that steroid treatment may contribute to the reduction in TNFa signaling observed in 2°BP patients.

Figure 6. Lower respiratory tract gene expression differs based on 2°BP status and is influenced by corticosteroid treatment. **b** Bar plot of GSEA analysis showing the Hallmark pathways that are downregulated in 2°BP patients. **c** Bar plot of GSEA analysis limited to steroid recipients showing the same Hallmark pathways as in **b**. The P-values in **b-e** were calculated using the fgsea package and Benjamini-Hochberg correction.

That said, we recognize that that our analyses and interpretation may not have been clear, and that our approach to assessing the impact of steroids needed to be more rigorous. As such, we have directly incorporated the reviewer’s feedback and now include several new analyses to assess the impact of steroids.

Moreover, we completely agree with the reviewer’s point that the high prevalence of steroid use in 2°BP patients makes assessing the contribution of steroids challenging. Thus, we have complemented the ‘steroid-only’ analyses described above with the following more in depth analyses that allow us to evaluate the impact of steroid exposure, including in 2°BP patients alone:

1. A new analysis evaluating the impact of days of steroid exposure prior to sampling on gene expression in 2°BP and No-BP patients. This analysis demonstrates that exposure to steroids in both 2°BP (new Fig. 6D) *and* No-BP patients (new Fig. 6E) is associated with suppression of several immune pathways important for antibacterial defense, including TNF α signaling. These results support the above-described hypothesis that corticosteroid exposure may predispose to 2°BP by suppressing immune signaling pathways important for antibacterial defense.

Figure 6. d Bar plot of GSEA analysis demonstrating Hallmark pathways associated with days of corticosteroid treatment in 2°BP patients. **e** Bar plot of GSEA analysis demonstrating Hallmark pathways associated with days of corticosteroid treatment in No-BP patients.

We now describe these findings and clarify the potential influence of steroids on our findings in the text as follows:

Line 329: “We sought to further understand the impact of steroid exposure on our transcriptomic findings by asking whether days of steroid treatment prior to sampling influenced gene expression. As expected, we found that days of steroid exposure correlated with suppression of several immune signaling pathways, including TNF α signaling, in both the 2°BP (Fig. 6d, Supp. Data 7) and No-BP (Fig. 6e, Supp. Data 8) groups. Together, these results suggested that a state of impaired immune defense, influenced by corticosteroid treatment, may exist in COVID-19 patients who develop 2°BP.”

2. A second new analysis focused on understanding the impact of corticosteroid exposure on the (inverse) relationship between bacterial mass and expression of host immune genes/pathways.

- We first asked whether restricting to just steroid-treated 2°BP patients (n=25) affects this relationship, but find that it does not (new Fig. 7f).
- We also repeated the analysis in No-BP patients (n=29) and observed similar findings at the pathway level (Fig. 7e).
- We then evaluate only the No-BP patients who received steroids (n=19, Fig. 7g), and again pathway results remained similar.
- Finally, we evaluate the No-BP patients who *did not* receive steroids (n=10, Fig. 7h) and again pathway results remained similar.

Figure 7. Lower respiratory tract immune gene expression inversely correlates with bacterial mass. **a** Volcano plots of genes that are associated with bacterial mass in 2°BP patients. **b** Scatter plots showing the relationship between *HLA-DRA* and *C1QC* gene expression and bacterial RNA mass in 2°BP patients. The black lines indicate the linear regression fit, and the ribbons indicate the 95% confidence interval of the fits. **c** Volcano plots of genes associated with bacterial mass in No-BP patients. **d-e** Bar plot showing Hallmark pathways associated with bacterial mass in **d** all 2°BP patients and **e** all No-BP patients. **f-g** Hallmark pathways associated with bacterial mass in **f** 2°BP and **g** No-BP patients, but limited to only steroid recipients. **h** Hallmark pathways associated with bacterial mass in the 10 No-BP patients who did not receive steroids prior to sample collection. The P-values in **a** and **c** were calculated using linear modeling (limma package) and Benjamini-Hochberg correction. The P-values in **d-h** were calculated using the fgsea package and Benjamini-Hochberg correction.

We were not able to specifically evaluate the 2°BP patients who *did not* receive steroids prior to sampling, as this group was so small (2 patients) that analyses would not be meaningful.

Taken together, our findings suggest that the relationship between greater bacterial mass and lower expression of immune signaling genes is not fully explained by steroid exposure. We now note this in the text as follows:

Line 354: “Using bacterial RNA mass as a continuous variable, differential expression analysis identified 4784 significant genes (FDR < 0.1) (Fig. 7a, Supp. Data 9), and revealed an inverse relationship between bacterial mass and the expression of several innate and adaptive immunity genes (e.g., *HLA-DRB1*, *C1QC*) (Fig. 7b). A similar differential expression analysis carried out in No-BP patients, who had lower bacterial RNA mass (Fig. 2a), yielded 30 significant genes (Fig. 7c, Supp. Data 10).

Gene set enrichment analysis (GSEA) confirmed a marked inverse relationship between bacterial RNA mass and immune signaling pathways important for antibacterial defense (e.g., TNF α , IL-6, IL-2) in both 2°BP (Fig. 7d, Supp. Data 11) and No-BP patients (Fig 7e, Supp. Data 12). Restricting analyses to only 2°BP or No-BP patients who had received corticosteroids (Figs. 7f, g, and Supp. Data 13 and 14) did not markedly change the relationship between bacterial mass and suppressed immune signaling, nor did restricting to the 10 No-BP patients without steroid exposure (Fig. 7h). Taken together, these results demonstrated that greater bacterial RNA mass was associated with lower expression of innate and adaptive immunity genes in mechanically ventilated COVID-19 patients, regardless of steroid exposure.

Please also see the response to reviewer 2, comment 5 regarding additional analyses that we have carried out to assess the impact of steroid exposure on our microbiome findings.

5. The microbiome is not fully analyzed here—just alpha diversity and one NMDS plot with a non-significant p-value. The conclusion [lines 332-333] that “2BP further disrupts microbial community structure in patients with existing SARS-CoV-2 infection” is not supported by the NMDS plot nor is it fully explored.

We have directly incorporated this feedback highlighting the need to provide a more comprehensive analyses of the lung microbiome. We note that we initially carried out three primary microbiome analyses: bacterial microbiome RNA mass/load, alpha diversity and beta diversity- assessed by Bray Curtis index.

We have now added additional analyses including:

- Comparative analysis of beta diversity in 2°BP versus No-BP using the Jaccard index (Supp. Fig. 2). Jaccard considers taxon presence/absence, in contrast to Bray Curtis, which considers taxonomic abundance.

Supplementary Figure 2. Beta diversity comparing tracheal aspirate samples from 2°BP versus No-BP patients, by Jaccard index. Non-metric multidimensional scaling (NMDS) plot demonstrating compositional differences in the lung microbiome of 2°BP (red dots) versus No-BP (blue dots) patients based on Jaccard dissimilarity index and the PERMANOVA test with 1000 permutations.

- Differential abundance analyses of taxa in 2°BP versus No-BP patients (Fig 2h), as well as a secondary analysis restricted to patients who received steroids (Supp. Fig. 5).

- Microbiome compositional analyses analyzing the concordance between NS and TA samples in 2°BP and No-BP patients (Fig 4d) by Spearman's correlation.
 - Please see response to comment 3 above.

4. Beta diversity analyses comparing Bray Curtis dissimilarity indices between nasal swab and tracheal aspirate samples in 2°BP and No-BP patients.

Supplementary Figure 15. Beta diversity comparing nasal swab to tracheal aspirate samples. Non-metric multidimensional scaling (NMDS) plots demonstrating upper versus lower airway beta diversity based on Bray-Curtis dissimilarity index between nasal swabs (NS, purple) and tracheal aspirate (TA, red), accounting for intra-participant clustering. **a** 2°BP patients and **b** No-BP patients. P values based on the PERMANOVA test with 1000 permutations.

5. Functional microbiome analyses examining bacterial metabolic pathways differentially abundant in 2°BP versus No-BP tracheal aspirate samples (Fig S7).

Supplementary Figure 7. Functional analysis of bacterial metabolic pathways based on 2°BP status. Metabolic pathways differentially abundant in 2°BP versus No-BP are highlighted. All pathways detected at false discovery rate < 0.25 are included in plot.

6. There is a small literature on microbiome concordance between the lower airways and nasal swabs. It would be helpful to briefly discuss this within the context of your metatranscriptomic results. Notably, there is much more concordance between the oral microbiome (another non-invasive surrogate site) and the lower airways than there appears to be with the nasal microbiome.

We appreciate this suggestion and now discuss our results in the context of prior literature assessing the concordance between the upper and lower airway microbiome as follows:

Line 420: “We found that 2°BP pathogens were not only detectable in the lower airway, but also in the upper airway in more than half of the cases, including prior to clinical pneumonia diagnosis. More broadly, however, we found only low-moderate correlation between the nasal and lower respiratory tract microbiomes, and did not observe any differences based on 2°BP status. A prior study using 16S rRNA gene sequencing to profile the microbiome of paired nasal and lower respiratory tract samples from children found significant correlations in taxonomic abundance between sites¹. In contrast, a recent metatranscriptomic study of mechanically ventilated COVID-19 patients reported significant differences in the microbial composition of the upper and lower airways², as did a 16S study comparing the lower airway microbiome to that of the oropharynx and upper respiratory tract⁴⁶. Additional studies are needed to more comprehensively understand relationships between the upper and lower airway microbiome, and how they differ based on sequencing technique (16S, metagenomics and metatranscriptomics) or on the presence of bacterial pneumonia.”

7. Lines 364-367: “Our data suggest that screening for other clinically important AMR genes may prove useful for identifying patients with drug-resistant pneumonia, with implications for both antimicrobial stewardship and infection control.” This is intriguing and may have significant clinical implications in the future. However, I would urge caution about over-interpretation of these results. In what percent of patients were the AMR genes identified in the NS samples of clinical relevance (i.e, found in both the lower airways and also in the organism causing the pneumonia)? What are the positive and negative predictive values of AMR genes found in TA samples? For instance, it would be unfortunate if a patient received a very broad antibiotic as a result of the NS AMR findings, when in fact their clinical pneumonia was due to different organism which did not harbor the AMR gene. The example of *mecA* nares screening is more nuanced than is suggested here in the discussion. Screening for MRSA is done in the nares because 1) the nares are a recognized reservoir of *S. aureus* (whereas many gram negative causes of pneumonia are not found in the nose) and 2) *mecA* screening is used for its negative predictive value, and not its positive predictive value.

We appreciate this point and have directly incorporated this feedback as follows:

To address the questions, “In what percent of patients were the AMR genes identified in the NS samples of clinical relevance (i.e., found in both the lower airways and also in the organism causing the pneumonia)?” and “What are the positive and negative predictive values of AMR genes found in TA samples?” we have done the following:

1. We have assessed the concordance between NS and TA samples for 14 pathogen-associated ARGs in a new Supp. Table 2c.

Supplemental Table 2c. Concordance of AMR gene detection in tracheal aspirate and nasal swab samples. Detection of 14 pathogen-associated AMR genes in tracheal aspirate or paired nasal swab samples is tabulated, as co-detection of the AMR gene in both sample types.

	Concordance of AMR gene detection in tracheal aspirate and nasal swab samples				
	Total patients with any gene detection	AMR gene detected at both sites	AMR gene detected only in TA	AMR gene detected only in NS	No available NS samples
ACT-MIR	2	0 (0.0%)	1 (50.0%)	0 (0.0%)	1 (50.0%)
AmpC	6	1 (14.3%)	2 (28.6%)	0 (0.0%)	4 (57.1%)
CTX-M	3	1 (33.3%)	0 (0.0%)	0 (0.0%)	2 (66.7%)
FONA-1	3	1 (33.3%)	0 (0.0%)	0 (0.0%)	2 (66.7%)
MAL-CKO	1	0 (0.0%)	0 (0.0%)	0 (0.0%)	1 (100.0%)
OXA-1	2	0 (0.0%)	0 (0.0%)	0 (0.0%)	2 (100.0%)
OXA-50	5	2 (40.0%)	0 (0.0%)	0 (0.0%)	3 (60.0%)
OXY	3	1 (33.3%)	0 (0.0%)	0 (0.0%)	2 (66.7%)
RAHN-1	3	1 (33.3%)	0 (0.0%)	0 (0.0%)	2 (66.7%)
SED-1	3	1 (33.3%)	0 (0.0%)	0 (0.0%)	2 (66.7%)
Qnr-1	1	1 (100.0%)	0 (0.0%)	0 (0.0%)	0 (0.0%)
MCR-1	1	1 (100.0%)	0 (0.0%)	0 (0.0%)	0 (0.0%)
Bla-Z	5	3 (60.0%)	2 (40.0%)	0 (0.0%)	0 (0.0%)
MecA	6	2 (33.3%)	2 (33.3%)	1 (16.7%)	1 (16.7%)

2. While this study was not designed to comprehensively and rigorously assess that performance of respiratory metagenomics to predict resistance phenotype from genotype, as a proof of principle, we have done so for *mecA* and *CTX-M*, two AMR genes with well-established genotype to phenotype relationships. We report results in Supp. Tables 2a and 2b. More specifically, we assessed:

- a. *mecA* and *Staphylococcus aureus* resistance to methicillin and related beta lactams
- b. *CTX-M* and *Enterobacteriaceae* resistance to ceftriaxone/3rd generation cephalosporins

Supplementary Table 2a. Relationship between *mecA* gene detection and susceptibility to methicillin in patients with cultured *S. aureus* isolates (N = 10). Detection of *mecA* in tracheal aspirate or nasal swab samples within -7/+7 days of 2°BP are compared with phenotypic methicillin/nafcillin/oxacillin susceptibility of *S. aureus* isolates cultured from tracheal aspirate. Two patients had *S. aureus* isolates for which phenotypic susceptibility was not performed (No data).

		Clinical Susceptibility			Calculated test performance			
		Methicillin-S (N = 6)	Methicillin-R (N = 2)	No data (N = 2)				
mecA detection in tracheal aspirates (TA)	mecA present (n, %)	2 (33.3%)	2 (100.0%)	0 (0.0%)	Sensitivity [CI]	1.00 [0.18-1.00]	Negative predictive value	1.00 [0.51-1.00]
	mecA absent (n, %)	4 (66.7%)	0 (0.0%)	2 (100.0%)	Specificity [CI]	0.67 [0.30-0.94]	Positive predictive value	0.5 [0.09-0.91]
	No TA samples available	0 (0.0%)	0 (0.0%)	0 (0.0%)				
mecA detection in nasal swabs (NS)	mecA present (n, %)	1 (16.7%)	2 (100.0%)	0 (0.0%)	Sensitivity [CI]	1.00 [0.18-1.00]	Negative predictive value	1.00 [0.51-1.00]
	mecA absent (n, %)	4 (66.7%)	0 (0.0%)	2 (100.0%)	Specificity [CI]	0.80 [0.38-0.99]	Positive predictive value	0.67 [0.12-0.98]
	No NS samples available	1 (16.7%)	0 (0.0%)	0 (0.0%)				

Supplementary Table 2b. Relationship between *CTX-M* (detection and phenotypic susceptibility to ceftriaxone in cultured *Enterobacteriaceae* isolates from patients with 2°BP. (N = 12 cultured isolates from 11 patients). Detection of *CTX-M* genes from tracheal aspirate or nasal swab samples within -7/+7 days of 2°BP were compared with phenotypic ceftriaxone susceptibility of *Enterobacteriaceae* isolates cultured from tracheal aspirate. One patient had an isolate for which phenotypic susceptibility was not performed (No data). Isolates with susceptibilities reported as “susceptible dose dependent” (N=1 pathogen-antibiotic combination) or “intermediate” (N=2 pathogen-antibiotic combinations) were considered as resistant for the purposes of this analysis.

		Clinical Susceptibility			Calculated test performance			
		Ceftriaxone-S (N = 5)	Ceftriaxone-R (N = 6)	No data (N = 1)				
CTX-M gene detection in tracheal aspirates (TA)	Present (n, %)	0 (0.0%)	4 (66.7%)	1 (100.0%)	Sensitivity [CI]	0.67 [0.30-0.94]	Negative predictive value	0.71 [0.36-0.95]
	Absent (n, %)	5 (100.0%)	2 (33.3%)	0 (0.0%)	Specificity [CI]	1.00 [0.57-1.00]	Positive predictive value	1.00 [0.51-1.00]
	No TA samples available	0 (0.0%)	0 (0.0%)	0 (0.0%)				
CTX-M gene detection in nasal swabs (NS)	Present (n, %)	0 (0.0%)	2 (33.3%)	0 (0.0%)	Sensitivity [CI]	1.00 [0.051-1.00]	Negative predictive value	1.00 [0.18-1.00]
	Absent (n, %)	1 (20.0%)	0 (0.0%)	1 (100.0%)	Specificity [CI]	1.00 [0.18-1.00]	Positive predictive value	1.00 [0.051-1.00]
	No NS samples available	4 (80.0%)	4 (66.7%)	0 (0.0%)				

We describe these findings in the results as follows:

Line 297: “We additionally performed an exploratory genotype-to-phenotype analysis focusing on *mecA* and resistance to methicillin (and related beta lactams) in *S. aureus*, and *CTX-M* and resistance to ceftriaxone in *Enterobacteriaceae* isolates. We found that detection of *mecA* in TA within 7 days of 2°BP clinical diagnosis had a sensitivity of 100% (95% CI 18-100%), specificity of 67% (95% CI 30-94%), negative predictive value (NPV) of 100% (95% CI 51-100%) and PPV of 50% (95% CI 9-91%) for MRSA (Supp. Table 2a). Similar results were found with NS samples, which had a NPV of 100% (95% CI 51-100%) and a PPV of 67% (95% CI 12-98%) for MRSA pneumonia. With respect to ceftriaxone resistance in *Enterobacteriaceae*, we found that detection of the ESBL gene *CTX-M* in TA had a sensitivity of 67% (95% CI 30-94%), specificity of 100% (95% CI 57-100%), NPV of 71% (95% CI 36-95%) and PPV of 100% (95% CI 51-100%) (Supp. Table 2b). In nasal swab samples, *CTX-M* had a NPV of 100% (95% CI 18-100%) and PPV of 100% (95% CI 5.1-100%) for ceftriaxone resistance. We found that detection concordance between NS and TA samples varied based on AMR gene and patient (Supp. Table 2c). Finally, we tested for associations between specific AMR genes and in-hospital mortality. No significant associations were identified, although the small sample size for each unique gene limited the power of this analysis (Supp. Table 2d).”

We also address them in the discussion as follows:

Line 444: “In several cases, pathogen-associated AMR genes were detectable before the clinical diagnosis of 2°BP, and were also found in the upper airway. Our genotype to phenotype assessment suggested that metatranscriptomic detection of clinically important AMR genes in NS or TA specimens may have utility for identifying patients with drug-resistant pneumonia. For instance, *mecA* detection in either TA or NS within 7 days of 2°BP clinical diagnosis had excellent sensitivity (100%) and NPV, and moderate specificity (67-80%) and PPV, for culture-confirmed MRSA pneumonia. In comparison, a meta-analysis of 22 studies found that *mecA* PCR had a pooled sensitivity and specificity of 70.9% and 90.3%, respectively⁴. We also found that *CTX-M* detection in either NS or TA had excellent PPV and moderate NPV for ceftriaxone resistance in *Enterobacteriaceae*. While these findings are promising and may have implications for both antimicrobial stewardship and infection control, a larger sample size is needed to draw robust conclusions.”

8. A significant limitation of the manuscript is the difficulty understanding which comparisons are being made and discussed, especially when steroid use confounds 2BP diagnosis. For instance, when evaluating the effects of steroid use, is the analysis limited to only those with 2BP? Or are all samples included regardless pneumonia diagnosis? The comparisons made in figure 6c seem to exclude individuals who did not receive steroids, however that doesn't account for the fact that steroids use is significantly confounded by 2BP diagnosis (and vice versa).

We appreciate the reviewer bringing this to our attention and have now made a concerted effort to clarify the details of each comparison in the figure panel titles, legend and in the text. We

also describe each of the analyses carried out to evaluate the potential impact of steroids on host gene expression analyses above in the response to comment 4.

9. Tables S2-S8 are missing from the supplementary materials file. I have questions about the data contained in them (specifically Tables S5 and S6 as described above in #4 and #8). Without these tables or additional info in the main text (described above) I'm unable to thoroughly review the manuscript.

We confirmed that our supplementary materials were uploaded with the initial submission so we suspect a technical issue in accessing them may have occurred. Regardless, we now confirm that supplementary materials have been uploaded properly.

Minor concerns:

10. Line 64: pathogens and the lung microbiome are both mentioned as though the two are separate entities. Do the authors wish to argue that pathogens can be analyzed separately from the lung microbiome? This sentence also asserts that there is a dynamic relationship between microbiome and immune response, but that this dynamic relationship has not been studied in much depth. I suggest re-phrasing this to acknowledge that one cannot assert a relationship is dynamic unless it has been adequately studied.

We appreciate this point. In sum, we believe that pathogens are best analyzed in the context of the greater lung microbiome. However, given that historically the medical community has evaluated pathogens separately from the lung microbiome, and given that pathogens have unique virulence and thus clinical implications, it can be reasonable to also assess them independently. Regardless, we have directly incorporated the feedback to re-phrase as suggested.

Line 94: "Despite their interconnected roles, few studies have assessed both the lower respiratory tract microbiome and host immune responses in critically ill patients, and none for the explicit purpose of studying post-viral 2°BP. A recent elegant study showcased how lower respiratory metatranscriptomics can effectively identify connections between host and microbial factors with clinical outcomes in COVID-19², however it did not focus on clinically confirmed 2°BP. Two recent diagnostic test studies demonstrated the potential of respiratory metatranscriptomics to improve the detection of pathogens in COVID-19 patients with ventilator associated pneumonia^{5,6}, but did not evaluate biological features of 2°BP."

11. Tracheal aspirate (TA) should be defined at the first use in the manuscript.

We appreciate this point and now define TA at first use.

Line 118: "We collected tracheal aspirate (TA) and nasal swabs (NS) periodically following intubation, and performed metatranscriptomic sequencing (Fig. 1a)."

12. Line 104: "increased bacterial RNA load"—does this refer to rRNA gene sequences or RNA transcripts from bacteria?

We appreciate the opportunity to clarify that this refers to total bacterial RNA mass (both rRNA gene sequences and bacterial transcripts). We have now clarified this as follows:

Line 612: “Total bacterial RNA mass was calculated based on the ratio of total bacterial reads in each sample to total reads aligning to the External RNA Controls Consortium (ERCC) RNA mass standards spiked into each sample⁷ according to a previously established approach^{8,9} employing the following equation: bacterial mass = [ERCC input mass x bacterial reads] / [ERCC reads]. Total RNA mass in each sample was calculated using the same approach but substituting total reads sequenced for total bacterial reads.”

13. Figure 1b: please clarify if these culture results represent only tracheal aspirates, or if nasal swab cultures are represented here as well. If nasal swabs are represented here, how did the study team adjudicate discordant culture results (TA vs. NS)?

We would like to clarify that culture results were only based on tracheal aspirates, a sample type widely used for detecting lower airway bacterial pathogens in the setting of pneumonia. We have adjusted the legend of Figure 1 to better clarify this point.

14. Please edit to exclude scientific notation for p-values in the manuscript text (e.g., $P = 3.03e-5$).

We appreciate this point. We have revised Figure 6d and Supplementary Tables 1a and 1b. Values less than $P < 0.0001$ were represented as <0.0001 ; all other P values are given with two significant digits.

15. In Figure 3, patient 1114 does not illustrate the stated finding. Pseudomonas is the #1 ranked pathogen in all samples shown here, so it cannot be described as “expanding”. Additionally, it is overly simplistic to say that in patient 1231 initiation of antibiotics resulted in falling pathogen abundance, since the pattern of falling Escherichia abundance was already established before susceptible antibiotics were started.

We appreciate this point and agree that this patient did not demonstrate expansion, and so now highlight a different case (patient 18 under new numbering system) which does clearly represent expansion. As noted above, per R3’s suggestion, we have restructured Figure 3 to highlight exemplary cases of distinct pathogen trajectories (as opposed to every patient). Furthermore, as discussed above, Figure 3 now shows pathogen mass and normalized pathogen mass in addition to rank, and highlights the following pathogen trajectories:

- a. pathogen expansion in the lower airway prior to 2°BP diagnosis in the absence of any antibiotic treatment
- b. reduction in pathogen abundance following 2°BP diagnosis
- c. pathogen persistence as the most abundant microbe in the lower airway despite initiation of appropriate antimicrobial therapy
- d. pathogen persistence as the most abundant microbe in the lower airway following treatment with antibiotics to which the pathogen was resistant

Figure 3. Dynamics of 2°BP pathogen over time relative to the date of clinical diagnosis highlighting examples of unique pathogen trajectories. The top row includes plots of genus level 2°BP pathogen rank based on bacterial reads per million (rpM) in the lung microbiome for each of four pathogen trajectories (expansion, reduction, persistence and resistance). The middle row includes plots of pathogen mass. The bottom row consists of plots of pathogen mass normalized to the total RNA mass of sample. Days relative to 2°BP clinical diagnosis are plotted on the X axis. Days during which patient received antibiotics to which 2°BP pathogen was phenotypically susceptible (light grey bar) or resistant (dark grey bar) are plotted below. The patient ID and pathogen genus are listed above each plot. Open circles denote samples in which the pathogen was not detected by metatranscriptomics.

We agree with the reviewer that it is overly simplistic to attribute the decline in *E. coli* abundance in patient 1231 (now patient 23) to antibiotics alone, as it is possible that other factors, such as the patient's immune response, enabled some degree of pathogen clearance prior to the initiation of antibiotics. We have reworded the text to clarify that this case simply represents an example where pathogen abundance decreased over time.

16. Line 214, please correct the p value given here.

We apologize for this typographical error and have corrected the text as follows:

Line 228: “We noted that patients with *P. aeruginosa* infections were more likely to exhibit this impaired clearance phenotype compared to patients with other types of 2°BP (P = 0.020, Supp. Fig. 11).”

17. Figure 4D. My version of the manuscript has random icons in place of the x- and y-axis labels.

We have ensured that this issue is now corrected in the longitudinal trajectory data previously found in Fig. 4D, and now located in Supp. Figure 1.

18. Figure 3 and 5, patient 1154 with *K. aerogenes* and an inducible ampC beta lactamase treated with pip-tazo: I’d argue that this individual did not receive an antibiotic to which his or her organism was susceptible. (See question 2.2 in <https://www.idsociety.org/practice-guideline/amr-guidance/#AmpC%CE%B2-Lactamase-ProducingEnterobacterales>).

We appreciate the reviewers’ point that piperacillin-tazobactam is not the optimal treatment for *ampC*-expressing organisms such as *K. aerogenes*. We have therefore revised the Methods as follows:

Line 504: “Lab-reported susceptibilities as “susceptible dose dependent” or “intermediate” were included as “resistant” given the possibility of clinical treatment failure. *Enterobacteriaceae* known to harbor inducible *ampC*-class genes (e.g., *Enterobacter* species, *Klebsiella aerogenes*, *Citrobacter* species¹⁰), were considered resistant to 1st-3rd generation cephalosporins and piperacillin-tazobactam regardless of phenotypic susceptibility.”

We have also changed the antibiotic profile from susceptible to resistant for patient 1154 (now patient 13) with *Klebsiella aerogenes*.

19. I would consider changing the patient numbering system in the manuscript. Patient 1145 and 1154 both have *Klebsiella* 2BPs and are heavily discussed here. The use of such similar numbers in the figures and text may lead to unnecessary confusion.

We appreciate this point and have changed the patient numbering system throughout the manuscript, figures, and supplemental files.

20. Line 278: The exact number of differentially expressed genes is missing.

We apologize for this typographical error and have corrected the text as follows:

Line 324: “This analysis yielded 425 differentially expressed genes (FDR < 0.1) (Supp. Data 5), but no evidence of suppressed TNF α signaling on GSEA, supporting the idea that corticosteroid treatment may have contributed to the suppressed immune signaling observed in patients who developed 2°BP (Fig. 6c, Supp. Data 6).”

21. Lines 279-280: I’m unclear how NOT finding suppressed TNF α in those with 2BP and

steroid use supports the hypothesis that corticosteroid treatment might contribute to the suppressed TNF α signaling in 2BP.

In this 'steroid-only' analysis, we attempted to assess differences in host response between 2BP and No-BP patients when differential steroid treatment was removed as a variable. This was done because 25/27 2BP patients received steroids while only 19/29 No-BP patients received steroids at the matched timepoint analysis utilized. One interpretation of our results (presented in Fig. 6c) is that without the imbalance in steroid treatment favoring 2BP patients, TNF α signaling is no longer downregulated in 2BP versus No-BP, suggesting but not proving that steroid treatment may contribute to the reduction in TNF α signaling observed in 2BP patients.

That said, as discussed above in response to comment 4, we have carried out additional more rigorous analyses to examine the impact of corticosteroids on immune signaling in 2BP and No-BP patients. These analyses, presented in Figs. 6d and 6e, demonstrate that steroid exposure is associated with suppression of several key innate immune signaling pathways including TNF α , in both 2BP and No-BP patients.

22. I highly commend the authors for using such a rigorous definition of pneumonia.

We appreciate this.

23. Discussion [lines 320-329]: While I agree that metatranscriptomics has the potential to enable early detection of 2BP, I do not think the data presented here provide strong evidence of this potential. Please temper expectations in this paragraph to ensure that readers understand the limited predictive capability of the NS transcriptomics data presented here.

We have incorporated this feedback to temper expectations, and have revised as follows:

Line 404: "Further studies in a larger, appropriately designed cohort, will be needed to ascertain the diagnostic performance of metatranscriptomics for 2BP."

Reviewer #2 (Remarks to the Author):

In this manuscript, the authors evaluate the upper and lower microbiome of critical COVID-19 patients undergoing mechanical ventilation through longitudinal sampling. The patients in this study are divided in two groups: those having secondary bacterial pneumonia (2BP), as determined by clinical evaluation and microbial culture, and those not presenting it. They evaluate the metatranscriptomics potential in detecting secondary pathogens in patients with 2BP, and determine the dynamics of these pathogens, as well as potential interactions between the microbiota and the host in these samples. One of their major claims is that there is a link between corticosteroid treatments, host gene expression pathways, and bacterial mass in the respiratory airways, which could lead to 2BP.

The manuscript presents promising results, but as a reviewer there are some issues that I would like to raise before article acceptance.

1- Given that metatranscriptomics analysis obtains the information from active bacterial/host transcripts, providing an analysis of bacterial functions in the samples analyzed, and whether they differ in 2BP and no-BP or with other covariates would be of utmost interest for the readers in the manuscript. Especially since the microbial taxonomic composition does not differ significantly between 2BP and no-BP, knowing whether or not there are significant functional shifts can bring novel insights into the understanding of 2BP in these patients.

We thank the reviewer for this suggestion and have now carried out a differential abundance analysis of microbial metabolic pathways in 2°BP versus No-BP using HUMAnN 3.0 (Beghini et al. eLife 2021;10:e65088), a leading software package for microbiome functional analyses. Using the recommended default settings of an FDR < 0.25, 28 metabolic pathways were upregulated in 2°BP. However, using our pre-established significance threshold of FDR < 0.1, no pathways were found to be significant.

This is presented in a new Supplementary Figure 7.

Supplementary Figure 7. Functional analysis of bacterial metabolic pathways based on 2°BP status. Metabolic pathways differentially abundant in 2°BP versus No-BP are highlighted. All pathways detected at false discovery rate < 0.25 are included in plot.

We now describe the approach to this analysis in the Methods and have added the following to the text:

Line 193: “Finally, we performed a functional analysis of bacterial metabolic pathways. No significant differences in metabolic pathway abundances were observed (FDR < 0.1) between 2°BP and No-BP patients (Supp. Fig. 7, Supp. Data 2).”

2- In line 177, when discussing the compositional differences between two groups based on a PERMANOVA analyses, the wording “approached statistical significance” is not appropriate. If the chosen cut-off to determine statistical significance is set to 0.05 as indicated in the methods, in this case the differences are not statistically significant and the wording in the results should not mislead the reader. In addition, the discussion indicates that one of the findings of the manuscript is that 2BP is characterized by a disruption of the lung microbiome, which is not shown by this analysis. Therefore, the claims from the manuscript should be modified accordingly.

We appreciate this point and have modified the text as follows. We have also included the results of a complementary beta diversity analysis based on Jaccard index:

Line 170: “Community composition of the lung microbiome in 2°BP versus No-BP patients did not differ based on either Bray Curtis index, which considers taxon abundance (Fig. 2d, P = 0.06 by PERMANOVA), or Jaccard distance, which considers taxon presence/absence (Supp. Fig. 2, P = 0.09).”

We would like to kindly note that we initially carried out three primary microbiome analyses: bacterial microbiome RNA mass/load, alpha diversity and beta diversity- assessed by Bray Curtis index. Of these, bacterial mass and alpha diversity (when measured at the time point at which the 2°BP pathogen was most abundant in the airway microbiome) were significantly different in 2°BP patients (Figure 2). Additionally, our assessment of the longitudinal dynamics of alpha diversity demonstrated differences in terms of the X intercept in 2°BP versus No-BP patients (now Supp. Fig. 1).

We have also done several additional microbiome analyses (described in response to R1, point 5; one of which was a differential taxonomic abundance analysis which demonstrated significant differences between 2°BP and No-BP (new Fig. 2H below).

Fig. 2h. Differentially abundant microbial genera in the lung microbiome of 2°BP versus No-BP patients.

Taken together, our results thus do indicate lung microbiome differences based on 2°BP status in some but not all analyses.

3- The code provided in the manuscript, available in a Github repository, is incomplete. In the README file, 3 *.R files are indicated to reproduce the analyses in the manuscript, but two of these files are missing, presumably removed in an update of the repository. Therefore, it is not possible to assess the analyses and the computational methods performed.

We apologize for this technical error with our Github repository, which we have now corrected.

4- Related to the issue above, the detection of AMR genes is not described in the methods. Which specific databases or functional annotations are used to identify these genes in the samples?

We appreciate the highlighting this oversight, which we have now corrected. We now describe the approach to AMR gene detection in the Methods:

Line 620: “Acquired ARGs annotated in the Antibiotic Resistance Gene-ANNOTation (ARG-ANNOT) database¹¹ were detected using the Short Read Sequence Typing (SRST2) algorithm¹². *TEM-1D*, ubiquitously present in library preparation reagents and thus in negative control water samples, was excluded. We subsequently evaluated all AMR genes detected with $\geq 5\%$ gene coverage or an average sequencing depth of ≥ 5 reads across the gene. We collapsed multi-mapped alleles for *mec*, *mcr*, *CTX-M*, *ampC* and *ampH* to the most abundant allele detected. Among patients with 2°BP, we described the longitudinal dynamics of pathogen-associated ARGs detected in tracheal aspirate samples collected 7 days before to 7 days after the 2°BP diagnosis. Gram-positive pathogen-associated ARGs included *BlaZ* and *mecA*; gram-negative pathogen-associated ARGs included *ACT-MIR*, *AmpC*, *CTX-M*, *FONA-1*, *MAL-CKO*, *OXA-1*, *OXA-50*, *OXY*, *RAHN-1*, *SED-1*, *Qnr-S*, *Mcr1* and *Mcr3*.

Also, determination of SARS-CoV-2 viral load is not reported.

We appreciate the opportunity to clarify that SARS-CoV-2 abundance/viral load was determined based on metatranscriptomics, and more specifically, *Betacoronavirus* reads per million mapping to the NCBI NT database using the CZ ID pipeline. We have clarified this in the Methods as follows:

Line 576: “SARS-CoV-2 viral load measured as *Betacoronavirus* rpM was determined from background-corrected metatranscriptomic taxon counts generated from the CZ ID pipeline.”

5- More details in the microbiome analysis should be reported. Besides aggregation at the genus level, is there any normalization or transformation applied before the different analyses performed?

We appreciate this input and have now provided additional details on the microbiome analyses as follows:

Line 578: “We then calculated bacterial relative abundance (measured in rpM) and bacterial mass as described below in the “RNA mass calculations” section. We compared bacterial mass, alpha diversity, and beta diversity for tracheal aspirate samples from a single timepoint between patients with and without 2°BP. Alpha diversity was calculated from bacterial relative abundance based on Shannon diversity index, and beta diversity was assessed based on the Bray-Curtis dissimilarity index and the Jaccard distance, using

PERMANOVA to assess statistical significance, and using the R package *Vegan* (v.2.6.4)¹³. To account for possible confounding, we performed four additional alpha diversity analyses and four additional beta diversity analyses; each analysis included one of the following covariates: 1) days of steroid receipt, 2) days of mechanical ventilation, 3) SARS-CoV-2 reads per million, 4) bacterial mass. We also compared beta diversity between nasal and tracheal aspirate samples from a single timepoint while accounting for intra-participant clustering, stratified by 2°BP status. Beta diversity results were plotted using non-metric multidimensional scaling (NDMS). Differential abundance analysis comparing tracheal aspirate samples from a single timepoint between patients with and without 2°BP was performed using the R package *DESeq2* (v1.36.0)¹⁴ by assessing bacterial genera present in $\geq 30\%$ of the samples. To account for receipt of steroids as a possible confounder to the differential abundance analysis, we performed a sensitivity analysis limited to patients who received steroids. To minimize the potential for false positive results with *DESeq2*¹⁵, we required a Benjamini-Hochberg false discovery rate < 0.001 .”

Additionally and most importantly, no association between the microbial composition and potential confounding factors has been reported. Do the antibiotics, SARS-CoV-2 viral load, bacterial load, corticosteroid treatment or other factors influence the taxonomic composition in the samples evaluated? It is intriguing, since in the host differential expression analyses (2BP vs no-BP) viral load is included as a covariate. Additionally, the discussion states that patients developing 2BP were more likely to have received corticosteroids. A thorough analysis of potential confounders or variables affecting the composition (and the function, see comment 1) should be performed.

We appreciate this point and have addressed it as follows. Firstly, we examined differences in alpha diversity in 2°BP and No-BP patients, respectively, based on days of steroid exposure, SARS-CoV-2 viral load, bacterial mass, and days of mechanical ventilation (Supp. Fig. 3).

Supplementary Figure 3. Alpha diversity based on days of steroid receipt, days of mechanical ventilation, SARS-CoV-2 viral load and bacterial mass. Analysis performed comparing alpha diversity between patients with No-BP (left columns, blue, N = 29) and 2°BP (right columns, red, N = 27) as compared by patients who were **a** above or below median days of steroid receipt (2 days in No-BP, 7 days in 2°BP), **b** above or below median number of mechanical ventilation days (6 days in both groups), **c** above or below the median SARS-CoV-2 viral load (93.2 rpM in No-BP, 15.9 rpm in 2°BP), **d** above or below the median bacterial mass (5.1 pg in No-BP, 90.9 pg in 2°BP). Boxes show median and 25th-75th percentiles, with whiskers from min to max. P values calculated by Wilcoxon tests.

Additionally, we examined beta diversity by Bray-Curtis index adjusted for days on ventilator, days of steroid receipt, bacterial mass, and SARS-CoV-2 viral load (Supp. Fig. 4).

Supplementary Figure 4. Beta diversity comparing tracheal aspirate samples from 2°BP versus No-BP patients, adjusted for days of steroid receipt, days of mechanical ventilation, SARS-CoV-2 viral load or bacterial mass. Non-metric multidimensional scaling (NMDS) plots demonstrating beta diversity based on Bray-Curtis dissimilarity index, adjusted for **a** days of steroid receipt, **b** days of mechanical ventilation, **c** SARS-CoV-2 reads per million, **d** bacterial mass. P values based on the PERMANOVA test with 1000 permutations.

As all patients received antibiotics, and there was wide heterogeneity in the antibiotics used and in the duration of treatment for each drug, this cohort was not amenable to assessing the impact of antibiotic treatment, although we appreciate the suggestion.

We now discuss these analyses focused on evaluating potential confounders of our microbiome analyses as follows:

Line 173: “We considered that corticosteroid exposure, days of mechanical ventilation prior to sampling, SARS-CoV-2 viral load and bacterial mass could each be potential confounders or effect modifiers of our lung microbiome analyses. In sensitivity analyses, however, we did not observe significant differences when evaluating the impact of each of these variables on alpha diversity (Supp. Fig. 3). Similarly, our beta diversity findings remained globally unchanged when adjusting for days of steroid exposure prior to sampling, days of mechanical ventilation, SARS-CoV-2 viral load or bacterial mass (Supp. Fig. 4).”

6- A more thorough investigation between the interaction of bacterial mass, corticosteroids and host gene expression should be performed. In the case of the association of host gene expression with bacterial mass, this is only performed in 2BP patients. Is it possible to deconfound bacterial mass and corticosteroid treatment? Also, what is the association of host gene expression with bacterial mass in no-BP patients?

We appreciate this suggestion and have carried out a new analysis focused on understanding the impact of corticosteroid exposure on the (inverse) relationship between bacterial mass and expression of host immune genes/pathways.

- a. We first asked whether restricting to just steroid-treated 2°BP patients affected this relationship, and found that it did not (new Fig. 7f).
- b. We also repeated the analysis in No-BP patients and observed fewer differentially expressed genes (Fig. 7d) but similar findings at the pathway level (Fig. 7e).
- c. We then evaluated only the No-BP patients who received steroids (Fig. 7g), and again pathway results remained similar.
- d. Finally, we evaluated the No-BP patients who *did not* receive steroids and also observed a similar inverse relationship between bacterial mass and innate immune signaling (Fig. 7h).

Figure 7. Lower respiratory tract immune gene expression inversely correlates with bacterial mass.

a Volcano plots of genes that are associated with bacterial mass in 2°BP patients. **b** Scatter plots showing the relationship between *HLA-DRA* and *C1QC* gene expression and bacterial RNA mass in 2°BP patients. The black lines indicate the linear regression fit, and the ribbons indicate the 95% confidence interval of the fits. **c** Volcano plots of genes associated with bacterial mass in No-BP patients. **d-e** Bar plot showing Hallmark pathways associated with bacterial mass in **d** all 2°BP patients and **e** all No-BP patients. **f-g** Hallmark pathways associated with bacterial mass in **f** 2°BP and **g** No-BP patients, but limited to only steroid recipients. **h** Hallmark pathways associated with bacterial mass in the 10 No-BP patients who did not receive steroids prior to sample collection. The P-values in **a** and **c** were calculated using linear modeling (limma package) and Benjamini-Hochberg correction. The P-values in **d-h** were calculated using the fgsea package and Benjamini-Hochberg correction.

Taken together, our findings suggest that the relationship between bacterial mass and suppressed immune signaling is not fully explained by steroid exposure. We now note this in the text as follows:

Line 350: “Finally, given the higher bacterial mass observed in 2°BP patients (Fig. 2a), and prior mouse studies suggesting that altered host gene expression in the context of experimental influenza infection is associated with bacterial overgrowth^{26,33}, we further investigated connections between the airway transcriptome and microbiome in 2°BP patients by testing whether bacterial RNA mass correlated with host gene expression. Using bacterial RNA mass as a continuous variable, differential expression analysis identified 4784 significant genes (FDR < 0.1) (Fig. 7a, Supp. Data 9), and revealed an inverse relationship between bacterial mass and the expression of several innate and adaptive immunity genes (e.g., *HLA-DRB1*, *C1QC*) (Fig. 7b). A similar differential expression analysis carried out in No-BP patients, who had lower bacterial RNA mass (Fig. 2a), yielded 30 significant genes (Fig. 7c, Supp. Data 10).

Gene set enrichment analysis (GSEA) confirmed a marked inverse relationship between bacterial RNA mass and immune signaling pathways important for antibacterial defense (e.g., TNFa, IL-6, IL-2) in both 2°BP (Fig. 7d, Supp. Data 11) and No-BP patients (Fig 7e, Supp. Data 12). Restricting analyses to only 2°BP or No-BP patients who had received corticosteroids (Figs. 7f, g, and Supp. Data 13 and 14) did not markedly change the relationship between bacterial mass and suppressed immune signaling, nor did restricting to the 10 No-BP patients without steroid exposure (Fig. 7h). Taken together, these results demonstrated that greater bacterial RNA mass was associated with lower expression of innate and adaptive immunity genes in mechanically ventilated COVID-19 patients, regardless of steroid exposure.”

7- In the discussion, it is stated that this study includes the novel use of host-microbe metatranscriptomics to study secondary bacterial infections, but other previous works have also employed metatranscriptomics to identify potential secondary infections by oral bacteria (Sulaiman et al, Nat Microbiol 2021 as reported in the references of the manuscript; or Wang et al, Signal Transduction and Targeted Therapy 2023), or even single-cell transcriptomics data to identify bacterial pathogens and their associations with host cells (Llorens-Rico et al, Nat Communications 2021). Therefore, their novelty claim should be modified accordingly.

We appreciate this point and acknowledge the importance of these prior excellent studies. We had worded as such because none had used a comparable clinical definition of 2°BP to identify patients for inclusion (e.g., a positive bacterial culture from the lower respiratory tract was considered sufficient for 2°BP diagnosis). That said, we do not want to underrecognize these prior contributions or have any misleading statements and thus have reworded as follows:

Line 468: “Strengths of our work include the use of host/microbe metatranscriptomics to study secondary bacterial infections, rigorous clinical adjudication of 2°BP, longitudinal sampling, and measurement of bacterial RNA mass, a biomarker not previously evaluated in studies of pneumonia. Limitations include a relatively small sample size and incomplete longitudinal sampling for all patients, which reduced the number of patients with analyzable samples near the date of 2°BP onset, and the number of analyzable longitudinal samples. This likely limited our ability to detect microbiome and host transcriptional differences that may have existed between 2°BP patients and controls. Our findings require further validation in independent cohorts. While several public COVID-19 respiratory transcriptomic datasets exist, none include adjudication of 2°BP status using a rigorous and standardized definition.”

We have also incorporated the following text into the Discussion:

Line 388: “Bacterial superinfection is a well-established contributor to influenza mortality¹⁸, yet in COVID-19 its role in mortality has been less clear^{2,19,20}. We found that 2°BP affected 39% of mechanically ventilated patients in our multicenter cohort, and was strongly associated with mortality. Our results notably differed from two important prior studies^{2,20} which did not find clear links between secondary bacterial infection and mortality in COVID-19 patients. This discrepancy may be explained by our use of both an established case definition²¹ and microbiological criteria to define 2°BP, as opposed to simply requiring a positive bacterial respiratory culture^{2,20}. We also found that patients who developed 2°BP were more likely to have received corticosteroids during their hospitalizations, suggesting that the therapeutic benefit of corticosteroids may come at the expense of increased 2°BP risk.”

And

Line 426: “In contrast, a recent metatranscriptomic study of mechanically ventilated COVID-19 patients reported significant differences in the microbial composition of the upper and lower airways², as did a 16S study comparing the lower airway microbiome to that of the oropharynx and upper respiratory tract⁴⁶. Additional studies are needed to more comprehensively understand relationships between the upper and lower airway microbiome, and how they differ based on sequencing technique (16S, metagenomics and metatranscriptomics) or on the presence of bacterial pneumonia.”

8- Bacterial RNA load/mass calculations are not clear, since the ratio reads/mass ratio can be affected by factors such as cell viability and transcriptional activity. Additionally, the amount of ERCC spiked into each sample should be detailed.

We appreciate this point and the need to clarify the methods for bacterial RNA mass calculations, including the amount of ERCC spiked into each sample. We have done so as follows:

Line 612: “Total bacterial RNA mass was calculated based on the ratio of total bacterial reads in each sample to total reads aligning to the External RNA Controls Consortium (ERCC) RNA mass standards spiked into each sample⁷ according to a previously established approach^{8,9} employing the following equation: bacterial mass = [ERCC input mass x bacterial reads] / [ERCC reads]. Total RNA mass in each sample was calculated using the same approach but substituting total reads sequenced for total bacterial reads.”

While the transcriptional activity of bacteria could affect this calculation, because most bacterial reads detected by metatranscriptomics map to 16S or 23S rRNA, bacterial transcriptional differences would not be expected to markedly change this mass estimate. That said, we have made sure to clarify that this mass calculation is an estimate and refers to total bacterial RNA.

9- In figure 2G of the pdf, the color legend for the number of cases is blank, missing the information to interpret the plot.

We appreciate the reviewer noting this issue which may have happened during .pdf conversion. We have ensured that it is now corrected.

10- In figure 2E, patient 1154 shows the bacterial abundance for *E. coli*, but in figure 3, this patient is indicated to have a 2BP caused by *Klebsiella*, which is the correct organism?

We appreciate the reviewer identifying this discrepancy. We have realized that the *E. coli* data are in fact from patient 1145 (not 1154) and we inadvertently mislabeled the patient ID in the figure. This is now corrected. As pointed out by reviewer 3, the use of such similar patient identifiers such as 1154 and 1145 can cause confusion and indeed it did. To incorporate reviewer 3's input, we have re-labeled all patients with metatranscriptomic data using IDs from 1 to 56. The revised version of figure 2e has patients 12 and 21 corresponding to 1145 and 1204.

11- In line 228: "Among the 12 patients who had NS samples collected prior to the date of clinical 2°BP diagnosis, 8/9 (88.9%)", is it 12 patients or 9?

We apologize for this error and have corrected it in the text as follows:

Line 249: "Among the 9 patients who had NS samples collected prior to the date of clinical 2°BP diagnosis, 8/9 (88.9%) had at least one etiologic pathogen detected (Figs. S12-S14)."

12- Figure 4D has corrupt x- and y- axis labels as well as panel titles and bottom-right legend.

We appreciate the reviewer identifying this error and we have corrected it. Note that Figure D is now in S12-S14.

13- Supplementary figures and tables numbering should be revised. In line 259 the authors refer to Figure S2 but I assume it is figure S3? Also in line 415 the authors point to Table S5 but it should probably refer to table S2?

We appreciate the reviewer bringing up this point and we have now revised the numbering of the supplementary figures and tables (which include additional new figures and tables).

14- In line 278 the number of genes differentially expressed is missing ("4XX"). According to the table S5 it should be 421?

We apologize for this typographical error and have corrected the text as follows:

Line 324: "This analysis yielded 425 differentially expressed genes (FDR < 0.1) (Supp. Data 5), but no evidence of suppressed TNF α signaling on GSEA, supporting the idea that corticosteroid treatment may have contributed to the suppressed immune signaling observed in patients who developed 2°BP (Fig. 6c, Supp. Data 6).

15- The average (and range) of reads sequenced per library (total reads as well as reads assigned to bacteria) should be reported, for future users to be able to produce comparable results in similar samples.

We appreciate this point and now report these metrics in the text as follows:

Line 537: "This generated an average of 7.99×10^7 total reads per sample (range: $0.51 \times 10^7 - 15.00 \times 10^7$), and an average of 3.11×10^6 bacterial reads per sample (range $1.19 \times 10^3 - 24.13 \times 10^6$."

16- I noticed that references 51 and 54 are the same (duplicated), so a revision of the references is needed

We appreciate the reviewer highlighting this issue and have corrected it.

Reviewer #2 (Remarks on code availability):

The code provided in the manuscript, available in a Github repository, is incomplete. In the README file, 3 *.R files are indicated to reproduce the analyses in the manuscript, but two of these files are missing, presumably removed in an update of the repository. Therefore, it is not possible to assess the analyses and the computational methods performed. Besides updating the Github repository, a release version of the publication should be archived in a different repository (i.e. Zenodo).

We appreciate the reviewer for highlighting this issue which we have corrected. The Github repository is now updated with all files needed to reproduce every analysis in the manuscript. We will also plan to incorporate the reviewer's recommendation to archive a "release version" of the repository when all code and analyses in the manuscript are finalized.

Reviewer #3 (Remarks to the Author):

This article delves into the metagenomic transcriptomic analysis of tracheal aspirate samples from COVID-19 mechanically ventilated patients afflicted with secondary bacterial pneumonia (2BP).

It also offers insights from partial metagenomic transcriptomes of nasopharyngeal swabs. Furthermore, by integrating transcriptomic data from human samples obtained from tracheal aspirates, likely epithelial cells, it provides a nuanced exploration of the microbial dynamics and immune features in COVID-19 patients experiencing secondary bacterial pneumonia.

While the clinical data within the manuscript are meticulously detailed and professionally elucidated, the depth of data analysis and exploration appears somewhat restrained. Nonetheless, the longitudinal cohort comprising multiple specimens enriches the study's scientific value. Notably, the absence of comprehensive microbiome characterization regarding 2BP underscores the significance of this study in broadening our understanding of this condition.

We thank R3 for highlighting the detail and elucidation of our clinical data, as well as for highlighting that our data including longitudinal sampling address the prior literature gap in microbiome characterization of 2°BP.

Here are some refined suggestions and inquiries:

1. Could there be a notable contrast in the rate of decline between 2BP and non-2BP cases in Figure 2D, despite the observed reduction in diversity in both scenarios?

We appreciate this suggestion and have tested whether the rates of alpha diversity decline over time (linear regression slopes) in 2°BP and No-BP patients differ over time. While intercepts at Day 0 differed, slopes do not. This figure panel has been moved to Supp. Fig. 1 to accommodate additional new data.

Supplementary Figure 1. Changes in lower respiratory microbiome alpha diversity over time. Linear regression of all samples from 2°BP patients (red dots) and No-BP patients (blue dots). Slope and Y-intercept are shown with 95% confidence intervals.

2. The classification of the small panels depicting individual patient statuses in Figure 3 into categories such as "pathogen expansion," "pathogen resistance," "decrease in pathogen abundance," and "most persistent" would enhance clarity. Representative samples from each category could be highlighted, with additional data relegated to the appendix.

We appreciate this suggestion and have reconfigured Figure 3 to highlight exemplary longitudinal plots of pathogen rank, mass and normalized mass for the following pathogen trajectory types:

- pathogen expansion in the lower airway prior to 2°BP diagnosis in the absence of any antibiotic treatment
- pathogen abundance reduction following 2°BP diagnosis
- pathogen persistence as the most abundant microbe in the lower airway despite initiation of appropriate antimicrobial therapy
- pathogen persistence as the most abundant microbe in the lower airway following treatment with antibiotics to which the pathogen was resistant

Figure 3. Dynamics of 2°BP pathogen over time relative to the date of clinical diagnosis highlighting examples of unique pathogen trajectories. The top row includes plots of genus level 2°BP pathogen rank based on bacterial reads per million (rpm) in the lung microbiome for each of four pathogen trajectories

(expansion, reduction, persistence and resistance). The middle row includes plots of pathogen mass. The bottom row consists of plots of pathogen mass normalized to the total RNA mass of sample. Days relative to 2°BP clinical diagnosis are plotted on the X axis. Days during which patient received antibiotics to which 2°BP pathogen was phenotypically susceptible (light grey bar) or resistant (dark grey bar) are plotted below. The patient ID and pathogen genus are listed above each plot. Open circles denote samples in which the pathogen was not detected by metatranscriptomics.

We have moved the complete plots for all tracheal aspirate samples (Supp. Figs 8-10) and nasal swab samples (Supp. Figs 12-14) to the supplementary materials.

3. Although the co-occurrence of secondary bacterial pneumonia with COVID-19 remains understudied, are there discernible parallels between the findings of this analysis and existing research on COVID-19 co-infections?

We thank the reviewer for this question, and now highlight both similarities and differences between our findings and studies examining secondary bacterial infections in COVID-19 patients, or examining the respiratory microbiome and host transcriptome in COVID-19 patients.

We comment on a markedly higher 2°BP-associated mortality found in our study versus two prior studies of COVID-19 secondary infections, although note that neither used an established clinical case definition of 2°BP and instead considered a positive bacterial culture equivalent to 2°BP.

Line 389: “We found that 2°BP affected 39.3% of mechanically ventilated patients in our multicenter cohort, and was strongly associated with mortality. Our results notably differed from two important prior studies^{2,20} which did not find clear links between secondary bacterial infection and mortality in COVID-19 patients. This discrepancy may be explained by our use of both an established case definition²¹ and microbiological criteria to define 2°BP, as opposed to only requiring a positive bacterial respiratory culture^{2,20}. We also found that patients who developed 2°BP were more likely to have received corticosteroids during their hospitalizations, suggesting that the therapeutic benefit of corticosteroids may come at the expense of increased 2°BP risk.”

We note that our findings of decreasing alpha diversity over time in line with those from Kitsios et al²², Llorens Rico et al.²³, and Sulaiman et al².

Line 406: “Lower respiratory tract infections are characterized by a loss of airway microbiome alpha diversity²⁴⁻²⁷, which is also observed over time in mechanically ventilated patients, including those with COVID-19^{22,23,28,29}.”

We also comment that our findings of distinct microbial communities in the upper versus lower airway are in line with those from Sulaiman et al². We additionally note that we did observe partial concordance in the detection of culture-confirmed 2°BP pathogens between the upper and lower airway, something which was not assessed in the Sulaiman et al. study:

Line 426: “In contrast, a recent metatranscriptomic study of mechanically ventilated COVID-19 patients reported significant differences in the microbial composition of the upper and lower airways², as did a 16S study comparing the lower airway microbiome to that of the oropharynx and upper respiratory tract⁴⁶.”

Line 420: “We found that 2°BP pathogens were not only detectable in the lower airway, but also in the upper airway in more than half of the cases, including prior to clinical pneumonia diagnosis.”

4. Given the meticulous documentation of antibiotic usage and resistance genes, could an integrated analysis further elucidate potential correlations between identified resistance genes and clinical outcomes? For instance, were any distinctive resistance gene patterns identified in cases involving *Pseudomonas aeruginosa* infection, as discussed?

We appreciate this suggestion and have carried out new correlation analyses between clinically relevant AMR genes and mortality. This is now provided in Supplementary Table 2d.

Supplementary Table 2d. Associations between AMR gene detection and in-hospital mortality.

Associations between detection of specific AMR genes in either nasal swab or tracheal aspirate metatranscriptomic data and in-hospital mortality were assessed by Fisher’s exact test.

Associations between AMR gene detection and in-hospital patient mortality			
Patients with gram negative 2°BP pathogens (N = 18)			
	Survivors (N = 7)	Deceased (N = 11)	
AMR gene	Prevalence	Prevalence	P value
ACT-MIR	0 (0.0%)	2 (18.2%)	0.50
AmpC	2 (28.6%)	5 (45.5%)	0.64
CTX-M	2 (28.6%)	1 (9.1%)	0.53
FONA-1	2 (28.6%)	1 (9.1%)	0.53
MAL-CKO	1 (14.3%)	(0.0%)	0.39
OXA-1	2 (28.6%)	(0.0%)	0.14
OXA-50	1 (14.3%)	4 (36.4%)	0.60
OXY	2 (28.6%)	1 (9.1%)	0.53
RAHN-1	2 (28.6%)	1 (9.1%)	0.53
SED-1	2 (28.6%)	1 (9.1%)	0.53
Qnr-1	0 (0.0%)	1 (9.1%)	1.00
MCR-1	0 (0.0%)	1 (9.1%)	1.00
Patients with gram positive 2°BP pathogens (N = 11)			
	Survivors (N = 6)	Deceased (N = 5)	
AMR gene	Prevalence	Prevalence	P value
Bla-Z	2 (33.3%)	3 (60.0%)	0.57
MecA	3 (50.0%)	3 (60.0%)	1.00

Line 310: “Finally, we tested for associations between specific AMR genes and in-hospital mortality, but found no significant associations (Supp. Table 2d).”

We also appreciate the suggestion to highlight distinctive resistance patterns in *Pseudomonas aeruginosa* and have done so in a new Supplemental Figure 17.

Patients with *Pseudomonas* 2°BP

Supplementary Figure 17. Antimicrobial resistance (AMR) genes detected in *Pseudomonas aeruginosa*. AMR genes detected in tracheal aspirate (TA) within 7 days of 2°BP clinical diagnosis grouped and colored by class.

AMR genes most common in patients with *Pseudomonas aeruginosa* included *OXA-50*, *CatB7* and *Aph3-IIb*, each detected in 5/6 (83%) of patients with 2°BP due to this organism. These genes confer resistance to beta-lactams, chloramphenicol and aminoglycosides, respectively.

We have added the following to the manuscript:

Line 281: “We noted that in patients with *P. aeruginosa* 2°BP, 83% had *OXA-50*, *CatB7* and *Aph3-IIb*, detected, which confer resistance to beta-lactams, chloramphenicol and aminoglycosides, respectively (Supp. Fig. 17).”

1. In the human transcriptomic analysis section of the results, the tacit assumption regarding the potential contribution of corticosteroid therapy lacks explicit clarification. Moreover, the sudden exploration of the "association between bacterial RNA load and host gene expression" without prior groundwork warrants elucidation. Additionally, while the method part delineates bacterial load calculation, specifying which samples were classified as high versus low load would enhance clarity.

We have recognized the need for more rigorous evaluation (and explanation of) the relationship between corticosteroid treatment and our host transcriptomic findings. We would like to note that we did initially attempt to evaluate the impact of steroids on our host gene expression analysis by performing the following:

1. Differential gene expression and gene set enrichment analysis (GSEA) between all 2°BP and No-BP patients, regardless of steroid status (Figures 6a-6b)
2. A secondary differential gene expression and GSEA between 2°BP and No-BP patients, restricted to only those patients who had received steroids prior to the time of sample collection (this reduced the comparison sample size by two 2°BP patients and 10 No-BP patients) (Fig. 6c).

In the 'steroid-only' analysis, (#2) we attempted to assess differences in host response between 2°BP and No-BP patients when differential steroid treatment was removed as a variable. This was done because 25/27 2°BP patients received steroids while only 19/29 No-BP patients received steroids at the matched timepoint analysis utilized. One interpretation of our results (presented in Fig. 6c) is that without the imbalance in steroid treatment favoring 2°BP patients, TNF α signaling is no longer downregulated in 2°BP versus No-BP, suggesting but not proving that steroid treatment may contribute to the reduction in TNF α signaling observed in 2°BP patients.

Figure 6. Lower respiratory tract gene expression differs based on 2°BP status and is influenced by corticosteroid treatment. **b** Bar plot of GSEA analysis showing the Hallmark pathways that are downregulated in 2°BP patients. **c** Bar plot of GSEA analysis limited to steroid recipients showing the same Hallmark pathways as in **b**. The P-values in **b-d** were calculated using the fgsea package and Benjamini-Hochberg correction.

That said, we recognize that that our analyses and interpretation may not have been clear, and that our approach to assessing the impact of steroids needed to be more rigorous. As such, we have directly incorporated the reviewer's feedback and now include several new analyses to assess the impact of steroids.

Moreover, we completely agree with the reviewer’s point that the high prevalence of steroid use in 2°BP patients makes assessing the contribution of steroids challenging. Thus, we have attempted to complement the ‘steroid-only’ analyses described above by the following more in depth analyses that allow us to evaluate the impact of steroid exposure, including in 2°BP patients alone:

2. A new analysis evaluating the impact of days of steroid exposure prior to sampling on gene expression in 2°BP and No-BP patients. This analysis demonstrates that exposure to steroids in both 2°BP (new Fig. 6d) *and* No-BP patients (new Fig. 6e) is associated with suppression of several immune pathways important for antibacterial defense, including TNF α signaling. These results support the above-described hypothesis that corticosteroid exposure may predispose to 2°BP by suppressing immune signaling pathways important for antibacterial defense.

Figure 6. d Bar plot of GSEA analysis demonstrating Hallmark pathways associated with days of corticosteroid treatment in 2°BP patients. **e** Bar plot of GSEA analysis demonstrating Hallmark pathways associated with days of corticosteroid treatment in No-BP patients.

We now describe these findings and clarify the potential influence of steroids on our findings in the text as follows:

Line 329: “We sought to further understand the impact of steroid exposure on our transcriptomic findings by asking whether days of steroid treatment prior to sampling influenced gene expression. As expected, we found that days of steroid exposure correlated with suppression of several immune signaling pathways, including TNF α , in both the 2°BP (Fig. 6d, Supp. Data 7) and No-BP (Fig. 6e, Supp. Data 8) groups. Together, these results suggested that a state of impaired immune defense, influenced by corticosteroid treatment, may exist in COVID-19 patients who develop 2°BP.”

We also appreciate R3 highlighting the need to provide context and rationale for the analysis between bacterial mass and host gene expression. We have now done so as follows:

Line 350: “Finally, given the higher bacterial mass observed in 2°BP patients (Fig. 2a), and prior mouse studies suggesting that altered host gene expression in the context of experimental influenza virus infection is associated with bacterial overgrowth^{16,17}, we further investigated connections between the airway transcriptome and microbiome in

2°BP patients by testing whether bacterial RNA mass correlated with host gene expression.”

With respect to which patients were classified as having high versus low bacterial load, we note that bacterial load was treated as a continuous variable for the analyses in Figure 7. We have clarified this with additional text in the Results and the Methods:

Line 354: “Using bacterial RNA mass as a continuous variable, differential expression analysis identified 4784 significant genes (FDR < 0.1) (Fig. 7a, Supp. Data 9).”

Line 651: “For the analyses of host gene expression and bacterial mass, we modelled gene expression on log₁₀-transformed bacterial mass (without adjusting for any other covariates).”

In addition, in the new Supp. Fig. 3, we now highlight the median bacterial RNA mass values of 5.1 pg in No-BP and 181.9 pg in 2°BP, and describe differences in alpha diversity based on patients above or below this median value of bacterial RNA mass.

Supplementary Figure 3. Alpha diversity based on days of steroid receipt, days of mechanical ventilation, SARS-CoV-2 viral load and bacterial mass. Analysis performed comparing alpha diversity between patients with No-BP (left columns, blue, N = 29) and 2°BP (right columns, red, N = 27) as compared by patients who were **a** above or below median days of steroid receipt (2 days in No-BP, 7 days in 2°BP), **b** above or below median number of mechanical ventilation days (6 days in both groups), **c** above or below the median SARS-CoV-2 viral load (93.2 rpM in No-BP, 15.9 rpm in 2°BP), **d** above or below the median bacterial mass (5.1 pg in No-BP, 90.9 pg in 2°BP). Boxes show median and 25th-75th percentiles, with whiskers from min to max. P values calculated by Wilcoxon tests.

The range of bacterial RNA mass values is also plotted in Fig. 2A.

Some minor issues warranting attention include:

1. Potential overlay of different images in Figure 4C.

We have confirmed that Figure 4C has been correctly rendered in the revision.

2. Ambiguity surrounding the mention of "4XX genes" in line 278.

We apologize for this typographical error and have corrected the text as follows:

Line 324: "This analysis yielded 425 differentially expressed genes (FDR < 0.1) (Supp. Data 5), but no evidence of suppressed TNF α signaling by GSEA, supporting the idea that corticosteroid treatment may have contributed to the suppressed immune signaling observed in patients who developed 2°BP (Fig. 6c, Supp. Data 6)."

Reviewer #3 (Remarks on code availability):

The code looks well-documented and complete, but I haven't conducted any testing on it.